

# Taxonomic analysis of Paraguayan samples of *Homonota fasciata* Duméril & Bibron (1836) with the revalidation of *Homonota horrida* Burmeister (1861) (Reptilia: Squamata: Phyllodactylidae) and the description of a new species

Pier Cacciali[1,2,3], Mariana Morando[4], Cintia D. Medina[4], Gunther Köhler[1], Martha Motte[5] and Luciano J. Avila[4]

[1] Herpetology Section, Senckenberg Forschungsinstitut und Naturmuseum, Frankfurt (M), Hesse, Germany
[2] Institute for Ecology, Evolution & Diversity, Biologicum, Johann Wolfgang Goethe Universität Frankfurt am Main, Frankfurt (M), Hesse, Germany
[3] Instituto de Investigación Biológica del Paraguay, Asuncion, Paraguay
[4] Grupo de Herpetología Patagónica. IPEEC-CENPAT-CONICET, Puerto Madryn, Chubut, Argentina
[5] Sección de Herpetologia, Museo Nacional de Historia Natural del Paraguay, San Lorenzo, Central, Paraguay

Corresponding author
Pier Cacciali,
pcacciali@senckenberg.de,
pier_cacciali@yahoo.com

## ABSTRACT

*Homonota* is a Neotropical genus of nocturnal lizards characterized by the following combination of characters: absence of femoral pores, infradigital lamellae not dilated, claws without sheath, inferior lamellae laterally not denticulate, and presence of a ceratobranchial groove. Currently the genus is composed of 10 species assembled in three groups: two groups with four species, and the *fasciata* group with only two species. Here, we analyzed genetic and morphologic data of samples of *Homonota fasciata* from Paraguay; according to Maximum Likelihood and Bayesian inference analyses, the Paraguay population represents an undescribed species. Additionally, morphological analysis of the holotype of *H. fasciata* (MNHN 6756) shows that it is morphologically different from the banded, large-scaled *Homonota* commonly referred to as "*H. fasciata*". Given the inconsistency between morphological characters of the name-bearing type of *H. fasciata* and the species commonly referred to as *H. fasciata*, we consider them as different taxa. Thus, *H. fasciata* is a *species inquirenda* which needs further studies, and we resurrect the name *H. horrida* for the banded, large-scaled *Homonota*. The undescribed species from Paraguay is similar to *H. horrida*, but can be differentiated by the high position of the auditory meatus relative to the mouth commissure (vs. low position in *H. horrida*); and less developed tubercles on the sides of the head, including a narrow area between the orbit and the auditory meatus covered with small granular scales with or without few tubercles (vs. several big tubercles on the sides of the head even in the area between the orbit and the auditory meatus). The new species is distributed in the Dry Chaco in South America. With the formal description of this species, the actual diversity of the genus *Homonota* is increased to 12 species. Furthermore, we infer phylogenetic relationships for 11 of the 12 described species of the genus, based on 11 molecular markers (two mitochondrial and nine nuclear genes), with concatenated and species tree approaches.

## INTRODUCTION

The genus *Homonota* is a gecko of Gondwanan origin, distributed in South America, being present in southern Bolivia, northern to southern Argentina, western Paraguay, Uruguay, and the Brazilian state of Rio Grande do Sul (*Gamble et al., 2008a*; *Morando et al., 2014*). Along its distribution it inhabits dry environments like Monte, Chaco, Espinal, Patagonian, Andean, and Pampas (*Morando et al., 2014*). Regardless of the ecoregion, the genus is terrestrial and with the exception of *Homonota fasciata*, all species have a reticulated coloration pattern that imitates lichens on rocky backgrounds (*Avila et al., 2012*: Fig. 1). Unlike other geckos in South America, *Homonota* is adapted to a terrestrial life-style being only infrequently found in trees (*Cei, 1986*).

All species in the genus are nocturnal, oviparous—laying one or two eggs—, insectivorous lizards that can be found frequently in human dwellings feeding on a wide range of arthropods (*Cei, 1986*; *Cei, 1993*; *Abdala, 1997*; *Carreira, Meneghel & Achaval, 2005*; *Ibargüengoytía & Casalinas, 2007*; *Kun et al., 2010*). Members of this genus are characterized by the following combination of characters: absence of femoral pores, infradigital lamellae not dilated, claws without sheath, inferior lamellae laterally not denticulate, and presence of a ceratobranchial groove (*Peters & Donoso-Barros, 1970*; *Cei, 1986*; *Carreira, Meneghel & Achaval, 2005*). Currently, ten species are recognized in this genus (*Cajade et al., 2013*), some of which have small distribution ranges restricted to one or few localities (e.g., *H. andicola*, *H. rupicola*, *H. taragui*, and *H. williamsii*) or medium-sized distributions of less than 400 km from north to south (e.g., *H. uruguayensis* and *H. whitii*), whereas others have wide distribution ranges (e.g., *H. borellii*, *H. fasciata*, *H. underwoodi*, and *H. darwinii*) (*Morando et al., 2014*). In fact, *H. darwinii* reaches 50°S latitude, the southernmost limit for the genus and for any gecko species of the world.

*Kluge (1964)* proposed a grouping arrangement for *Homonota*, in which he placed *H. borellii*, *H. fasciata*, *H. horrida*, and *H. uruguayensis* in one group, and *H. darwinii*, *H. underwoodi*, and *H. whitii* in another. But a recent molecular analysis carried out by *Morando et al. (2014)* shows a different arrangement dividing the genus into three groups (i.e., the *borellii*, *whitii*, and *fasciata* groups). This last group is the least diverse with only two species, whereas each of the former two contain four species (*Morando et al., 2014*). The two species belonging to the *fasciata* group are *H. underwoodi* described by *Kluge (1964)* and *H. fasciata* with a complex taxonomic history discussed by *Abdala & Lavilla (1993)*.

*Duméril & Bibron (1836)*, based on a single specimen from ''Martinique'', described *Gymnodactylus fasciatus*. *Burmeister (1861)* described *Gymnodactylus horridus* from Sierra del Challao, in Mendoza Province (Argentina). *Gray (1845)* erected the genus *Homonota* to accomodate the ''Guidichaud's [sic] Scaled Gecko'' *Gymnodactylus gaudichaudii Duméril & Bibron, 1836* (currently *Garthia gaudichaudii*), but according to *Vanzolini (1968)*, Gray actually used a specimen of *Homonota darwinii* (and not *G. gaudichaudii*), for the description of *Homonota*, so that *Homonota darwinii* is the actual type species of the

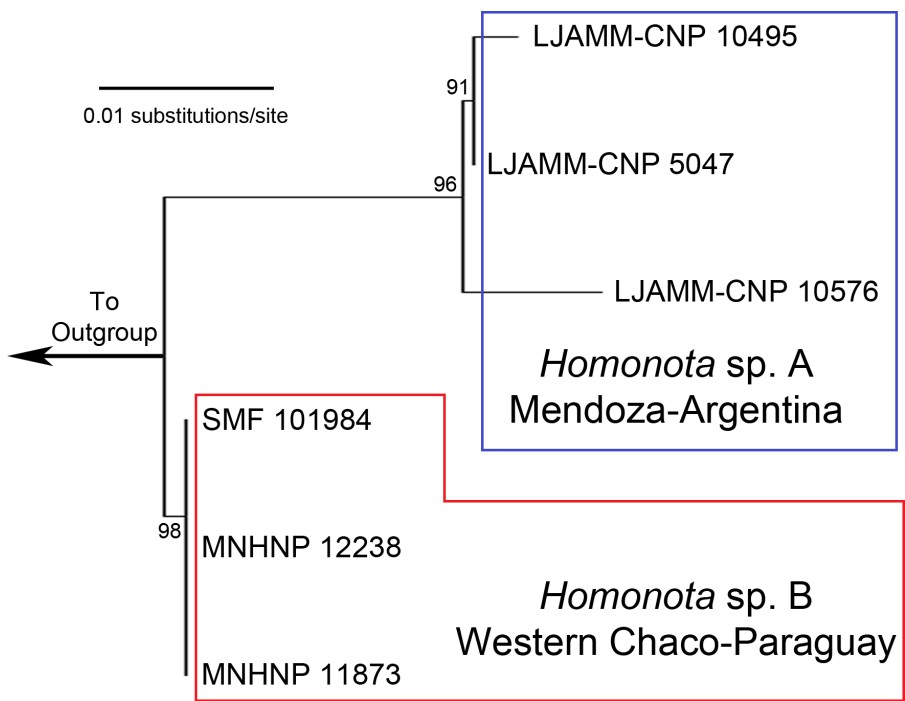

0.01 substitutions/site

**Figure 1** **Maximum Likelihood tree.** Maximum Likelihood clusters of *Homonota fasciata* from Argentina (blue square) and from Paraguay (red polygon), obtained from 16S mtDNA barcode sequences. Until name assignation, we refer to them as *Homonota* sp. A and *Homonota* sp. B respectively. Outgroup: *Phyllopezus przewalskii.*

genus. In a brief publication, *Berg (1895)* provided a description of a lizard he named *Gymnodactylus mattogrossensis* from Mato Grosso (Brazil, without any specific locality data), referring to a single specimen (not vouchered) given to him by his colleague Julio Koslowsky. *Kluge (1964)* moved these three names to the genus *Homonota* recognizing *H. horrida* and *H. fasciatus* [sic] as a valid species and transferring *Gymnodactylus mattogrossensis* to the synonymy of *H. horrida*. *Kluge (1964)* stated that these species are similar but differ in the number of interorbital scales (10–14 in *H. horrida* vs. 16 in the holotype of *H. fasciata*), the denticulation of ear opening (strongly denticulate all around the opening in *H. horrida* vs. a slight denticulation on the anterior margin in *H. fasciata*), size of postmental scales (moderately enlarged in *H. horrida* vs. greatly enlarged in *H. fasciata*), and size and shape of gular scales (large and plate-like in *H. horrida* vs. small and granular in *H. fasciata*). According to this author, *H. horrida* is present in southern Bolivia and Brazil, Paraguay, and northwestern Argentina, whereas the distribution of *H. fasciata* is unknown because its type locality "Martinique" is apparently based on a mistake, and no more additional locality records were available. *Abdala & Lavilla (1993)* suggested that diagnostic characters between *H. horrida* and *H. fasciata* as proposed by *Kluge (1964)* were intraspecific variation, and they synonymized *H. horrida* with *H. fasciata*. Since then the name *H. fasciata* was applied to the banded, large-scaled *Homonota* distributed from northern Paraguay and southern Bolivia, to Río Negro Province (central Argentina).

An almost complete molecular phylogenetic analysis was performed by *Morando et al. (2014)* including topotypes of all the recognized species. For *H. fasciata* the authors used specimens from Mendoza, since the original type locality (Martinique) is a mistake, and *Abdala & Lavilla (1993)* restricted the type locality of *H. fasciata* to Mendoza (in den Schluchten der Sierra bei Challao), which is actually the type locality for *Homonota horrida*.

In Paraguay, *Homonota fasciata* is distributed mainly in the Dry Chaco, with only one record in a transition zone of Dry Chaco with Humid Chaco (*Cacciali et al., 2016*). Given that *H. fasciata* has a complex taxonomic history, is one of the widest distributed members of the genus, and the almost complete absence of samples from Paraguay in previous publications, here we follow an integrative approach to assess the taxonomic status of samples from this country. First, within the framework of a barcoding project of Paraguayan herpetofauna, we generated molecular data and inferred a first round of hypotheses. Second, based on 11 genes, we inferred the taxonomic position of the Paraguayan populations in a phylogenetic tree that includes all the described species. Lastly, we analyzed detailed morphological data and also examined the holotype of *H. fasciata*.

## MATERIALS AND METHODS

### Genetic analyses

We carried out a first genetic inspection of the taxonomic status of Paraguayan populations currently referred to as *Homonota fasciata* using sequences of the mtDNA 16S gene as it was proved to be a useful tool for taxonomic identification (*Jansen & Schultze, 2012*; *Batista et al., 2014*; *Köhler, Vargas & Lotzkat, 2014*) with a desirable relation of cost/benefit. The Paraguayan samples ($N = 3$, GenBank accession numbers presented in Appendix S1, Supplementary Information online) from two localities were compared with available samples of the species from Mendoza, Argentina (used by *Morando et al., 2014*) located ~1.400 km in straight line ($N = 3$). Localities of vouchers used for genetic analyses are shown in Appendix S2. Paraguayan samples were collected with collecting permits SEAM No 04/11 and SEAM No 133/2015 issued by the Secretaría del Ambiente in Paraguay. Specimens were euthanized using anesthetic injections of barbituric acids (Tiopental Sódico® 1 g).

Tissue samples were first washed for 15 h with 50 µl Phosphate-buffered saline (PBS) (diluted of 1:9 PBS: $H_2O$). They were digested in a solution of Vertebrate lysis buffer (60 µl per sample) and proteinase K (6 µl per sample) at 56 °C for 15 h. Protocol for DNA extraction followed *Ivanova, Dewaard & Hebert (2006)*. After extraction, DNA was eluted in 50 µL Tris-EDTA (TE) buffer. Amplification of mtDNA 16S gene fragments was made using the eurofins MWG Operon primers L2510 (forward: 5′–CGCCTGTTTATCAAAAACAT–3′) and H3056 (reverse: 5′–CCGGTCTGAACTCAGATCACGT–3′) in an Eppendorf Mastercycler® pro. PCR conditions were: 94 °C–2 min, 40 ×[94 °C–35 s, 48.5 °C–35 s, 72 °C–1 min], 72 °C–10 min. Sequencing was performed using a BigDye® Terminator with the following cycling conditions: 95 °C–1 min, 30 ×[95 °C–10 s, 50 °C–10 s, 60 °C–2 min], with 10 µl of reaction volume.

The examination of chromatograms and generation of consensus sequences was performed using SeqTrace 0.9.0 (*Stucky, 2012*). Sequences were aligned first automatically with Clustal W (*Larkin et al., 2007*) followed by a visual inspection and edition if necessary, with the freeware MEGA 6 (*Tamura et al., 2013*). The alignment and the tree are available at TreeBase (ID: 20987). The substitution model for our dataset was identified according to the corrected (for finite sample size) Akaike Information Criterion (AICc) (*Burnham & Anderson, 2002*) and computed in MEGA 6.

We estimated the uncorrected genetic pairwise distances for our dataset, and ran Maximum Likelihood (ML) analysis with 30,000 bootstrap replicates in MEGA 6. We used *Phyllopezus przewalskii* as outgroup (SMF 100495, GenBank accession number MF278834), due to availability of relevant genetic information.

We used species delimitation methods to assess the degree of intraspecific divergences and to support the cluster arrangement suggested by the ML approach. This exploration was performed separately for the alignment and for the tree. The alignment was analyzed with ABGD (*Puillandre et al., 2012*) using simple distances to compare with the uncorrected genetic distance. For the tree based on 16S analysis, we applied the Poisson tree process (PTP) (*Zhang et al., 2013*) conducted through the bPTP web Server (http://species.h-its.org/), using default parameters and the outgroup removed. This algorithm does not require an ultrametric tree as input (*Zhang et al., 2013*), and it is a robust tool to estimate species delimitation from ML phylogenetic reconstructions (*Tang et al., 2014*). To assess the phylogenetic position of the Paraguayan samples within the genus, we used data from the recently published phylogenetic inference by *Morando et al. (2014)* and generated new sequences for all markers for samples from Paraguay (Appendix S3). We followed *Morando et al. (2014)* for amplification of the same two mitochondrial and nine nuclear genes, alignment protocols and gene and species trees approaches. Primers are specified in Appendix S4 .

Consensus sequences for each sample was generated with Sequencher v4.8 ([TM]Gene Codes Corporation Inc. 2007, Ann Arbor, MI, USA), and aligned with Mafft (*Katoh & Standley, 2013*). Confirmation of open reading frames for protein-coding genes was made by translation into amino acids.

The best evolutionary substitution model for each gene was selected using the AICc (*Burnham & Anderson, 2002*) and ran in jModelTest v2.1.10 (*Darriba et al., 2012*). Recombination was tested and excluded for nuclear genes using RDP: Recombination Detection Program v3.44 (*Martin & Rybicki, 2000*; *Heath et al., 2006*). We conducted Separate Bayesian analyses (BI) for each gene using MrBayes v3.2.2 (*Ronquist & Huelsenbeck, 2003*). Four heated Markov chains (with default heating values) and run for five million generations were used for each analysis. The equilibrium samples (after 25% of burn-in) were used to generate a 50% majority-rule consensus tree, and posterior probabilities (PP) were considered significant when ≥0.95 (*Huelsenbeck & Ronquist, 2001*). Maximum Likelihood (ML) analyses for each gene were performed with RAxML v7.0.4 (*Stamatakis, 2006*), based on 1,000 rapid bootstrap analyses for the best ML tree.

We performed concatenated analyses with ML and BI for the following datasets: (1) two mitochondrial genes combined, (2) nine nuclear genes combined, (3) all genes combined.

Likelihood analyses were performed using RAxML v7.0.4, based on 1,000 rapid bootstrap analyses. Bayesian analyses were conducted using MrBayes v3.2.2, with four heated Markov chains (using default heating values) and run for 50 million generations, with Markov chains sampled at intervals of 1,000 generations. Equilibrium samples (after 25% of burn-in) were used to generate a 50% majority-rule consensus tree, and posterior probabilities (PP) were considered significant when ≥0.95 (*Huelsenbeck & Ronquist, 2001*).

For construction of a species tree incorporating the multispecies coalescent approach, we used the hierarchical Bayesian model integrated in *Beast v1.8.0 (*Drummond & Rambaut, 2007*). For all genes were run two separate analyses for 100 million generations (sampled every 1,000 generations). Clades with PP >0.95 were considered strongly supported.

To ensure that convergence was reached before default program burn-in values, we evaluated convergence of Bayesian MCMC phylogenetic analyses (MrBayes and *Beast) by examining likelihood and parameter estimates over time in Tracer v1.6 (*Rambaut, Suchard & Drummond, 2009*). All parameters were between 157 and 23,400 effective sample sizes (ESS).

All alignments and trees were stored in TreeBase (ID: 20987); phylip files produced by RAxML were converted to nexus with ALTER (*Glez-Peña et al., 2010*), and trees merged with matrices in Mesquite v3.2 (*Madison & Madison, 2017*).

## Morphological approach

Voucher specimens are listed in Appendix S5. Coordinates are presented in decimal degrees and WGS 84 datum, and all the elevations are in meters above sea level (masl). Institution codes follow *Sabaj Pérez (2014)*.

Metric characters were taken following *Avila et al. (2012)*, and include snout-vent length (SVL) from tip of snout to vent; trunk length (TrL) distance from axilla to groin from posterior edge of forelimb insertion to anterior edge of hindlimb insertion; foot length (FL) from tip of claws of the 4th toe to heel; tibial length (TL) greatest length of tibia, from knee to heel; arm length (AL) from tip of claws of the 3rd finger to elbow; head length (HL) distance between anterior edge of auditory meatus and snout tip; head width (HW) taken at level of the temporal region; head height (HH) maximum height of head, at level of parietal area; eye-nostril distance (END) from the anterior edge of the eye to the posterior edge of the nostril; eye-snout distance (ESD) from the anterior edge of the eye to the tip of the snout; eye-meatus distance (EMD) from the posterior edge of the eye to the anterior border of the ear opening; interorbital distance (ID) interorbital shortest distance; internostril distance (IND). Meristic data consist of: number of keeled dorsal tubercles (DT) from occipital area to cloaca level; number of transversal rows of ventral scales (TVS), counted longitudinally at midline from the chest (shoulder level) to inguinal level; number of longitudinal rows of ventral scales (LVS), counted transversally at midbody; number of supralabial scales (SL); number of infralabial scales (IL); number of fourth toe lamellae (4TL); and number of third finger lamellae (3FL). Paired structures are presented in left/right order. In the color descriptions, the capitalized colors and the color codes (in parentheses) are those of *Köhler (2012)*.

Based on the genetic clusters recognized by the barcoding analysis, we performed a discriminant function analysis (DA). As a first step we tested normality with Shapiro–Wilk

**Table 1  Pairwise distances for 16S.** Uncorrected pairwise genetic distances (in percentages) based on 16S mtDNA among samples of Species A from Argentina (white cells) and Species B from Paraguay (gray cells) formerly referred as *H. fasciata*. Minimum and maximum values between species in bold.

|  | LJAMM-CNP 5047 | LJAMM-CNP 10495 | LJAMM-CNP 10576 | MNHNP 11873 | MNHNP 12238 | SMF 101984 |
|---|---|---|---|---|---|---|
| LJAMM-CNP 5047 | – |  |  |  |  |  |
| LJAMM-CNP 10495 | 0.4 | – |  |  |  |  |
| LJAMM-CNP 10576 | 0.6 | 1.0 | – |  |  |  |
| MNHNP 11873 | **1.8** | 2.0 | **2.5** | – |  |  |
| MNHNP 12238 | 2.0 | 2.2 | 2.4 | <0.01 | – |  |
| SMF 101984 | 2.0 | 2.2 | 2.4 | <0.01 | <0.01 | – |

**Table 2  Fixed sites in the alignment of 16S.** The 11 fixed sites differences on our 16S mtDNA alignment among three samples of Species A from Argentina (Ar) and three of Species B from Paraguay (Pa), formerly referred to as *Homonota fasciata*. The numbers indicate nucleotide position.

|  | 007 | 154 | 191 | 216 | 218 | 284 | 302 | 320 | 339 | 405 | 489 |
|---|---|---|---|---|---|---|---|---|---|---|---|
| Species A (Ar) | T | G | C | T | – | T | A | A | C | T | T |
| Species B (Pa) | C | A | – | C | R | C | C | C | T | C | C |

(*W*) test (*Shapiro, Wilk & Chen, 1968*; *Zar, 1999*). Then we performed the DA including variables with normal distribution, analyzing continuous characters (metrics) that are sensitive to ontogeny, separated from discrete (non-sensitive to body growth) characters. All statistical procedures were performed with Past 3.14 (*Hammer, Happer & Ryan, 2001*).

# RESULTS

## Phylogenetic inference

Following we present the size of each aligned gene (in brackets) and the best substitution model identified: 16S [527 bp]: GTR+G; 12S [951 bp]: GTR+G; cyt-b [794 bp]: TRN+I+G; MXRA5 [961 bp]: TPM1lf+G, NKTR [1074 bp]: TRN+G, SINCAIP [449 bp]: TPM2 lf+G, RBMX [600 bp]: HKY+G, DMXL1 [959 bp]: HKY+G, ACA4 [1218 bp]: HKY+G, PRLR [543 bp]: TRN+G, Homo_30b [664 bp]: TRN+I, Homo_19b [642 bp]: F81+G.

The ML tree based on an initial exploration with 16S mtDNA gene sequences shows two separate clades of geckos, formerly referred to as *Homonota fasciata* (Fig. 1), with uncorrected 16S p-distances ranging between 1.8 and 2.5% (Table 1). In the alignment we identified 11 fixed different sites between these clades (Table 2). We interpret the documented genetic differences as evidence for heterospecificity of these two clades. Thus, we recognize two potential species of geckos formerly referred to as *H. fasciata*: Species A (sampled in Low Monte ecoregion) and Species B (sampled in Dry Chaco, Paraguay).

The ABGD analysis for the 16S dataset resulted in the recognition of three groups (1- Species A, 2- Species B, 3- Outgroup) with a range of intraspecific genetic variation from 0.1 to 0.77%; and two groups (1- *Homonota*, 2- Outgroup) with an intraspecific variation of 1.29% (Appendix S6). This is only slightly higher than the higher intraspecific distance between two of our samples (*p*-distance=1.0% between LJAMM-CNP 10495 and LJAMM-CNP 10576; Table 1) of Species A, whereas the intraspecific distance among

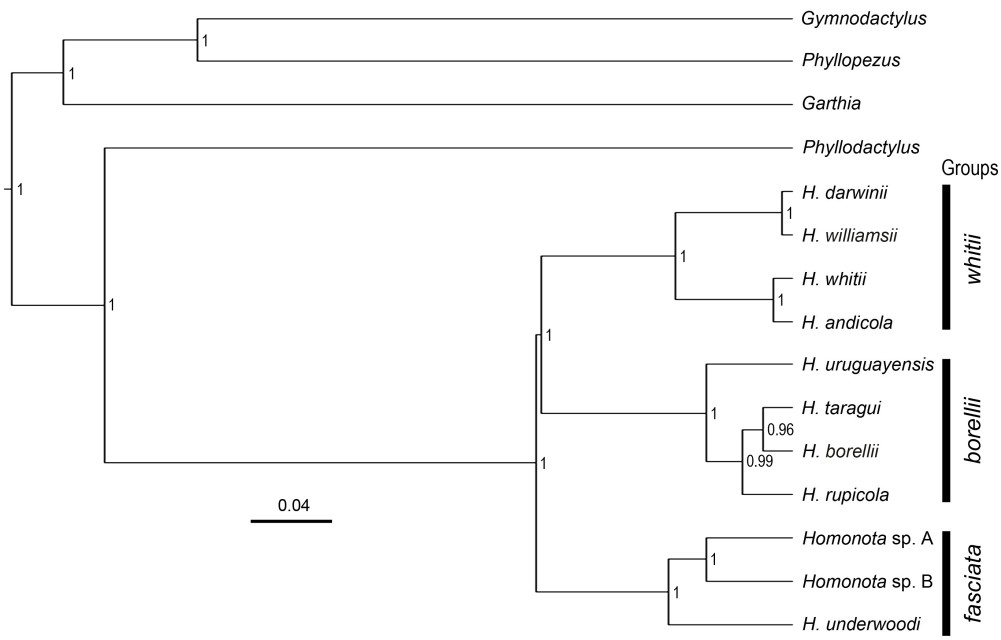

**Figure 2** **Species tree.** Species tree of *Homonota* and related taxa inferred with *Beast, showing the position of the two clades (*Homonota* sp. A and *Homonota* sp. B) formerly referred as *H. fasciata*. Bar represents substitutions per site. Only values ≥0.95 are shown.

specimens of Species B (<0.01%). The PTP also proposed two different clades (both with ML and Bayesian algorithms) grouping separately Argentinean samples (Species A) and Paraguayan samples (Species B) (Appendix S7). Species A was inferred as the sister taxon of Species B in nine of the 11 independent gene trees obtained with both BI and ML (Appendix S8). Exceptions include: 1-the gene Homo_30b (both with BI and ML), which infer Species B as sister of the clade Species A +*H. underwoodi*; 2-DMXL1 inferred the *borelli* group as sister to Species A+Species B (both with BI and ML); 3-the gene SINCAIP (ML only) showed the groups *fasciata* and *whitii* nested together; 4- the gene NKTR with ML inferred *H. underwoodi* as a member of a different group (Appendix S8).

All phylogenies inferred from concatenated datasets of (1) two mitochondrial genes combined, (2) nine nuclear genes combined, (3) all genes combined with both BI and ML showed high support in recognizing Species B from Paraguay as a sister to Species A from Argentina, with *Homonota underwoodi* as sister to these two within the *fasciata* group (Appendix S9). The species tree inferred with *Beast presents the same arrangement within the *fasciata* group as those inferred by BI and ML using concatenated datasets (Fig. 2).

## Morphological analyses

All the continuous variables had normal distributions, but two discrete variables (SL and IL) did not (Table 3), thus, they were excluded from further morphological analysis. Convex hulls for metric variables show significant discrimination between Species A and Species B, which support the cluster differentiation inferred from molecular data (Fig. 3). The most contributing variables were SVL and TrL for Axis 1 (Appendix S10). Sexual dimorphism was not recorded for Species A, whereas an evident sexual dimorphism in Species B was

**Table 3 Statistical values for mophological analyses.** Normality Shapiro–Wilk (W) values for metric (above) and meristic (below) characters showing the *p* value. Values shaded in gray did not reach normality. See Materials and Methods section for reference to the acronyms.

| | Continuous | | | | | | | | | | | | |
|---|---|---|---|---|---|---|---|---|---|---|---|---|
| | SVL | TrL | FL | TL | AL | HL | HW | HH | END | ESD | EMD | ID | IND |
| *W* | 0.976 | 0.969 | 0.955 | 0.986 | 0.987 | 0.960 | 0.954 | 0.961 | 0.975 | 0.965 | 0.971 | 0.979 | 0.952 |
| *p* | 0.604 | 0.377 | 0.377 | 0.902 | 0.949 | 0.223 | 0.126 | 0.282 | 0.602 | 0.314 | 0.471 | 0.688 | 0.113 |

| | Discrete | | | | | | |
|---|---|---|---|---|---|---|---|
| | DT | TVS | LVS | SL | IL | 4TL | 3FL |
| *W* | 0.956 | 0.956 | 0.967 | 0.798 | 0.705 | 0.943 | 0.955 |
| *p* | 0.138 | 0.153 | 0.349 | $9.61E^{-6}$ | $2.01E^{-7}$ | 0.064 | 0.126 |

documented (Fig. 3). Nevertheless, the probability ellipse (confidence = 95%) propose a high overlap, and females of Species B is the most different group (Fig. 3).

Regarding meristic data, sexual dimorphism is more pronounced in *H. fasciata* than in *Homonota* sp. "Paraguay" (Fig. 4). Raw data are available in Appendices S11 (metric variables) and S12 (meristic variables).

## Taxonomic implications

We take the significant level of genetic differentiation between these two clusters of *Homonota* as evidence for the recognition of two different taxa. In order to correctly assign names to these two species, we examined the relevant primary types of the nominal taxa in this species complex. The holotype of *H. fasciata* is MNHN 6756 (LSID: urn:lsid:zoobank.org:act:14CDAB98-810F-43B3-8F16-B29C830AB80C). As mentioned above, the original type locality of *H. fasciata* was given as "Martinique" and is without doubt erroneous. A detailed analysis of MNHN 6756 (Fig. 5) revealed that it differs in pholidosis in several significant characters from the biological species currently referred to as *H. fasciata* (Table 4), such as margin of auditory meatus (Fig. 6), size of first infralabial scale (Fig. 7), and the arrangement of dorsal scales (Fig. 8). Given these differences in several taxonomically important scalation traits, there is no doubt that MNHN 6756 is not conspecific with the biological species currently referred to as *H. fasciata*. The scalation traits of MNHN 6756 presented above resemble the external morphology of *Homonota uruguayensis* (Vaz-Ferreira & Sierra de Soriano, 1961). However, *H. uruguayensis* does not have transversal bands on the dorsum, and in the original description of *H. fasciata* transversal bands on the dorsum of the type specimen are mentioned. In its current state, the holotype of *H. fasciata* is completely bleached and does not show any trace of banding (Fig. 5). In conclusion, we cannot link the holotype of *H. fasciata* to any of the known populations of *Homonota* which renders this name a *species inquirenda* which needs further studies and cannot be linked to either Species A or Species B. Our examination of the lectotype of *H. horrida* (IZH-R 1) revealed that it is conspecific with our Species A which is supported by the fact that the Argentinian specimens used in our genetic analysis are from the general area of the type locality of *H. horrida*. We therefore resurrect it from synonymy with *H. fasciata* and apply it to our Species A. As mentioned above, the original
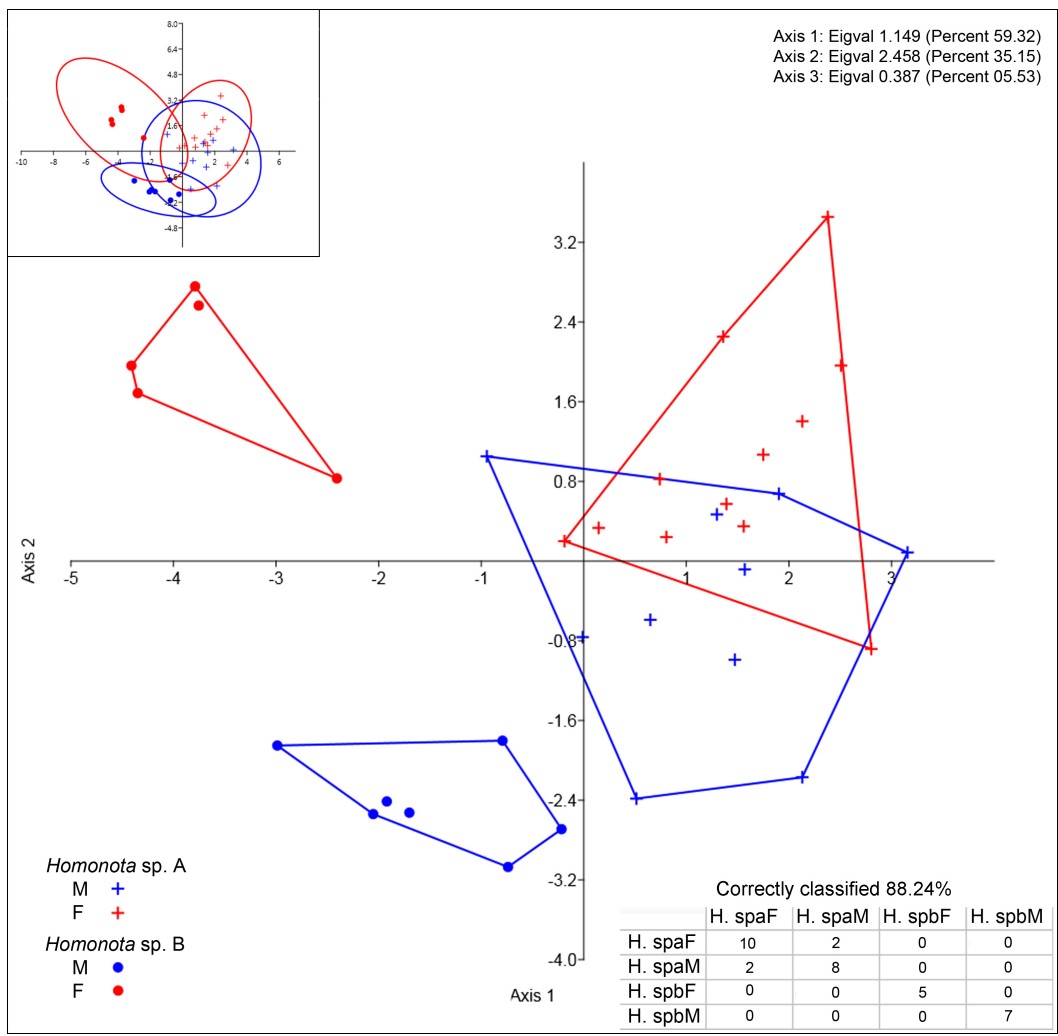

**Figure 3** **Discriminant analysis of continuous variables.** DA scatter plot of individual scores of the three most informative axes for continuous variables (See Appendix S10) of *Homonota* sp. A (Hspa in the table) and *Homonota* sp. B (Hspb in the table). Capital letters "F" and "M" refer to females and males respectively. Inset on upper left corner shows the 95% confidence intervals.

description of *H. mattogrossensis* is very brief, does not provide a precise type locality (and no representative of the genus *Honomota* is known to occur in Mato Grosso do Sul) and no type material or other voucher specimen is known. Therefore this name cannot be applied to any of the known populations of this genus and we consider *Homonota mattogrossensis* to constitute a *nomen dubium*.

No name is available for our Species B and we therefore describe it as a new species below, presenting also a species account and a redescription of *H. horrida*. The electronic version of this article in Portable Document Format (PDF) will represent a published work according to the International Commission on Zoological Nomenclature (ICZN), and hence the new names contained in the electronic version are effectively published under that Code from the electronic edition alone. This published work and the nomenclatural
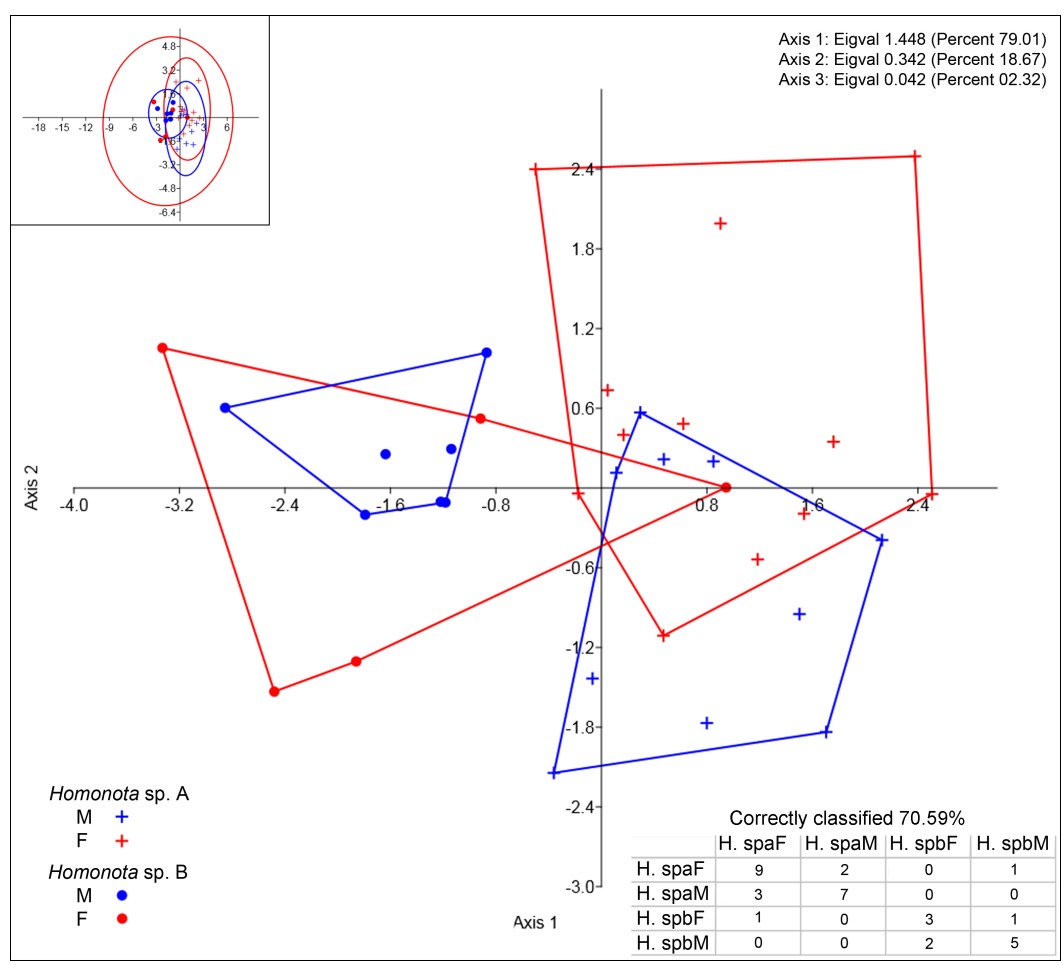

Axis 1: Eigval 1.448 (Percent 79.01)
Axis 2: Eigval 0.342 (Percent 18.67)
Axis 3: Eigval 0.042 (Percent 02.32)

*Homonota* sp. A
M +
F +
*Homonota* sp. B
M •
F •

Correctly classified 70.59%

|        | H. spaF | H. spaM | H. spbF | H. spbM |
|--------|---------|---------|---------|---------|
| H. spaF | 9 | 2 | 0 | 1 |
| H. spaM | 3 | 7 | 0 | 0 |
| H. spbF | 1 | 0 | 3 | 1 |
| H. spbM | 0 | 0 | 2 | 5 |

**Figure 4** **Discriminant analysis of discrete variables.** DA scatter plot of individual scores of the three most informative axes for discrete variables (See Appendix S10) of *Homonota* sp. A (Hspa in the table) and *Homonota* sp. B (Hspb in the table). Capital letters "F" and "M" refer to females and males respectively. Inset on upper left corner shows the 95% confidence intervals.

**Table 4** **Morphological differences.** Differences in morphological traits between MNHN 6756 (holotype of *Homonota fascia*) and *Homonota* sp. commonly referred as *H. fascia*.

| Trait | MNHN 6756 | *Homonota* sp. |
|-------|-----------|----------------|
| Margin of auditory meatus | Smooth | Strongly serrated |
| Enlarged tubercle on the auditory meatus | Absent | Present |
| Postmental scale | Exceptionally large | Almost same size of first infralabial |
| Dorsal scales | Small and widely spaced | Large and juxtaposed |

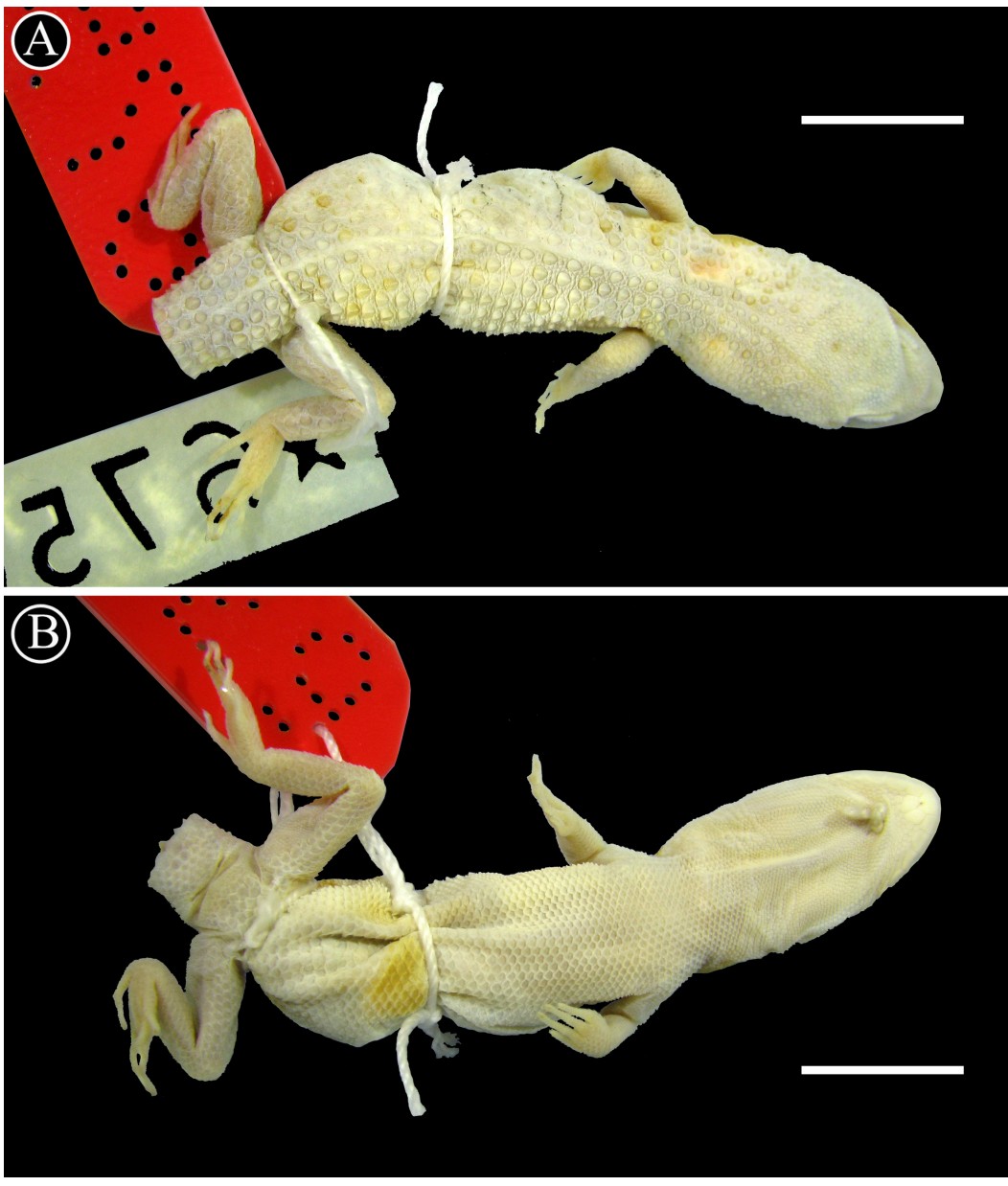

**Figure 5  Holotype of *Homonota fasciata*.** Dorsal (A) and ventral (B) views of the holotype of *Homonota fasciata* (MNHN 6756). Scale bar = 1 cm.

acts it contains have been registered in ZooBank, the online registration system for the ICZN. The ZooBank LSIDs (Life Science Identifiers) can be resolved and the associated information viewed through any standard web browser by appending the LSID to the prefix http://zoobank.org/. The LSID for this publication is: urn:lsid:zoobank.org:pub:7233E738-D8B3-424D-B1FC-7CA903BED5A0. The online version of this work is archived and available from the following digital repositories: PeerJ, PubMed Central and CLOCKSS.

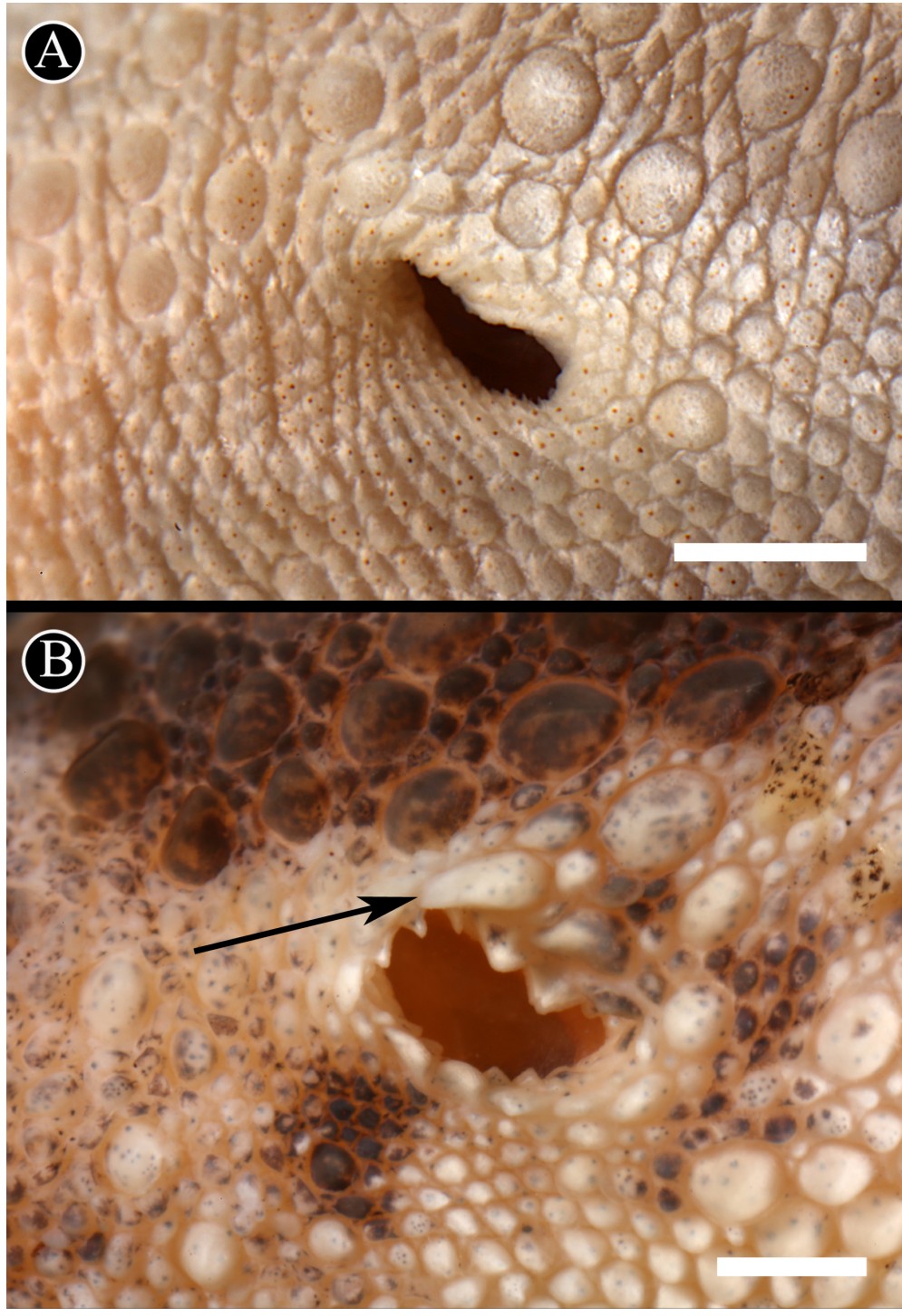

**Figure 6 Auditory meatus.** Detail of the auditory meatus of the holotype of *H. fasciata* (A) showing an even edge, and *Homonota* sp. (B) showing the serrate edge. Black arrow indicates an enlarged tubercle associated to the upper edge of the auditory meatus, absent in the holotype of *H. fasciata*. Head to the right. Scale bar = 1 mm.

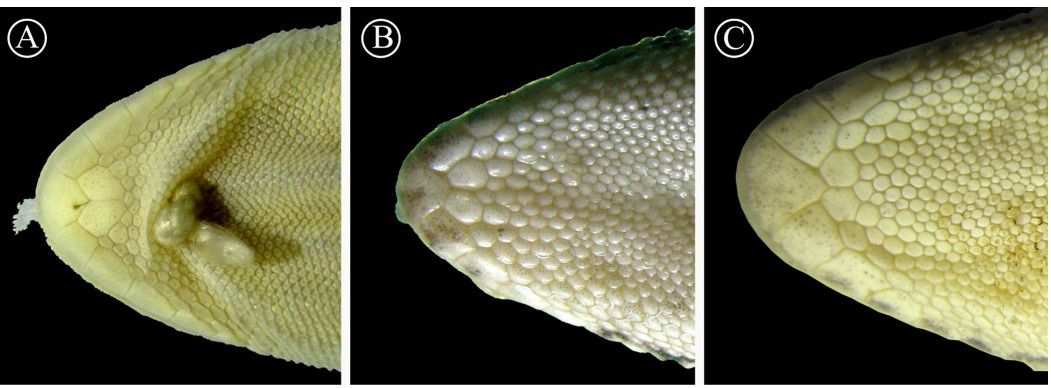

**Figure 7 Detailed view of postmental scales.** Detail of the mental region, showing the large size of the postmental scales of the holotype of *H. fasciata* (A), compared with *Homonota* sp. A (B) and *Homonota* sp. B (C). Vouchers: A- MNHN 6756; B- MNHNP 12238; C- LJAMM-CNP 6520.

***Homonota horrida*** (*Burmeister, 1861*) **sp. reval.**
  - *Gymnodactylus horridus Burmeister, 1861*
*Type locality*: "in den Schluchten der Sierra bei Challao", Mendoza, Argentina.
*Types*: Original description based on three syntypes. Lectotype (IZH-R 1, Fig. 9) and paralectotype (IZH-R 2) designation according to *Müller (1941)*.
  - *Wallsaurus horridus Underwood 1954*
  - *Gymnodactylus pasteuri Wermuth 1965*
  LSID: urn:lsid:zoobank.org:act:27FAE0B5-2E88-46C5-A296-F7BBE0B20AE6

*Diagnosis*: A large species of *Homonota* with a dark dorsal color (grey or brown) with a pattern of clear transversal bands connected with a vertebral stripe. Additionally, it is differentiated from any other *Homonota* by the large size and development of the keeled scales on the head (including laterals) and dorsum.

*Redescription of the lectotype* (Fig. 9): Adult male, SVL 44 mm, TrL 19 mm, tail 49 mm, FL 8.0 mm, TL 8.5 mm, AL 12.0 mm, HL 11.1 mm, HW 8.5 mm, HH 6.3 mm, END 3.7 mm, ESD 4.6 mm, EMD 4.1 mm, ID 4.3 mm, IND 1.4 mm; rostral wider than high; nares surrounded by rostral, supranasal, two postnasals, and first SL; SL 9/9; one elongated tubercular scale on the mouth commissure; upper region of the muzzle covered by big homogeneous juxtaposed scales; upper surface of the head covered with medium-sized (smaller than those on the muzzle) homogeneous juxtaposed scales intermixed with small granules; superciliary scales imbricated, associated to spiny-like scales on the posterior half of the orbit; lateral sides of the head heterogeneously covered profusely with large keeled tubercles and small granular (sometimes elongated) scales; auditory meatus oblique and with serrated edge, and one big scale on the upper border; IL 6/6; mental triangular; postmentals big (about twice the size of the following posterior scales) contacting the mental, the first IL, and a row of six posterior scales (the two centrals smaller); scales under the head reducing in size posteriorly; dorsolateral parts of the neck with granular juxtaposed scales mixed with tubercles; throat region covered by imbricated cycloid scales; dorsum covered with 16 strongly keeled scales separated by one or two

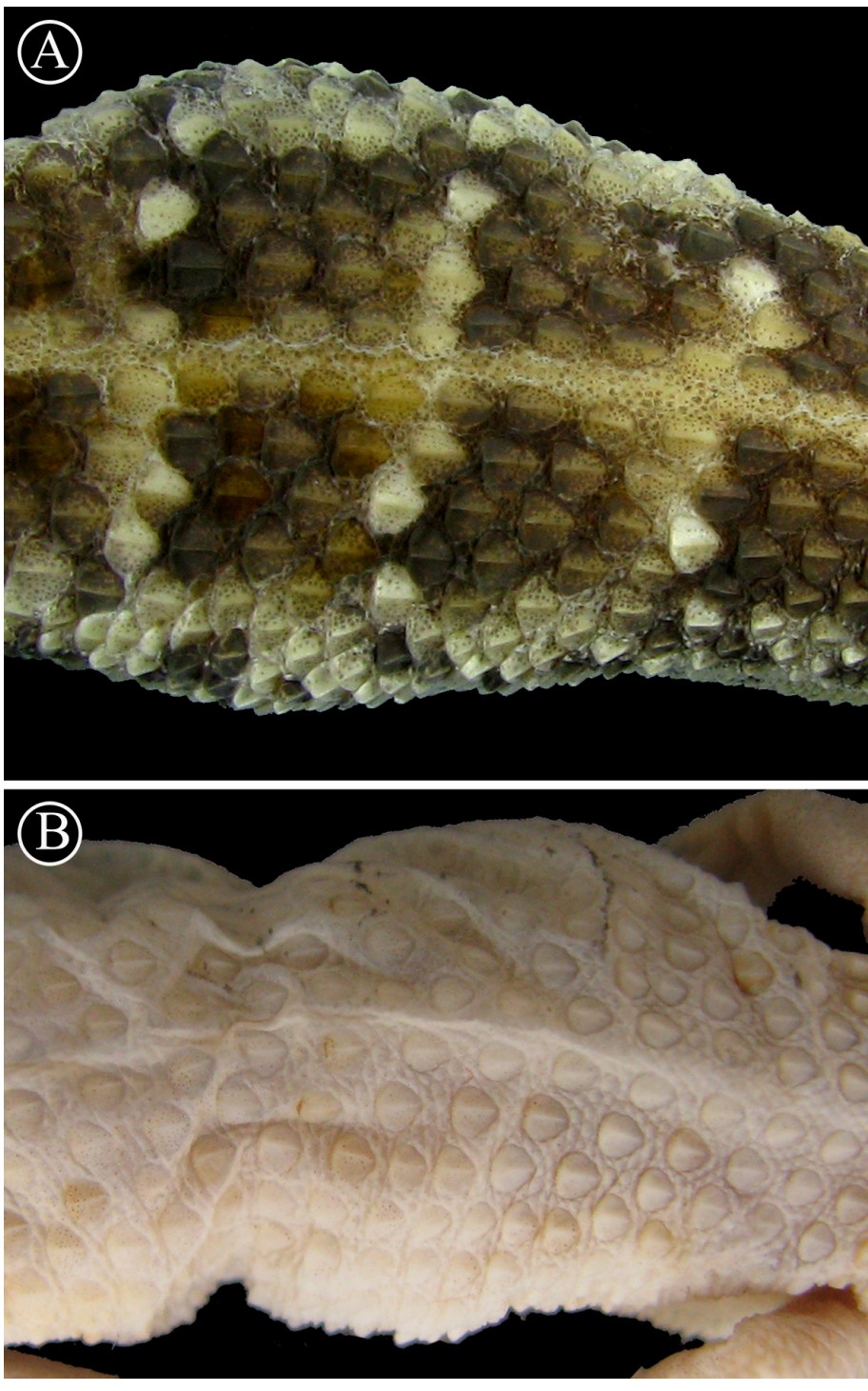

**Figure 8** **Detailed view of dorsal scales.** Lineal arrangement of dorsal scales of *Homonota* sp B.
(A) commonly referred to as *H. fasciata*, and the holotype of *H. fasciata* (B). Note the different pattern in
the squamation. Head to the right.

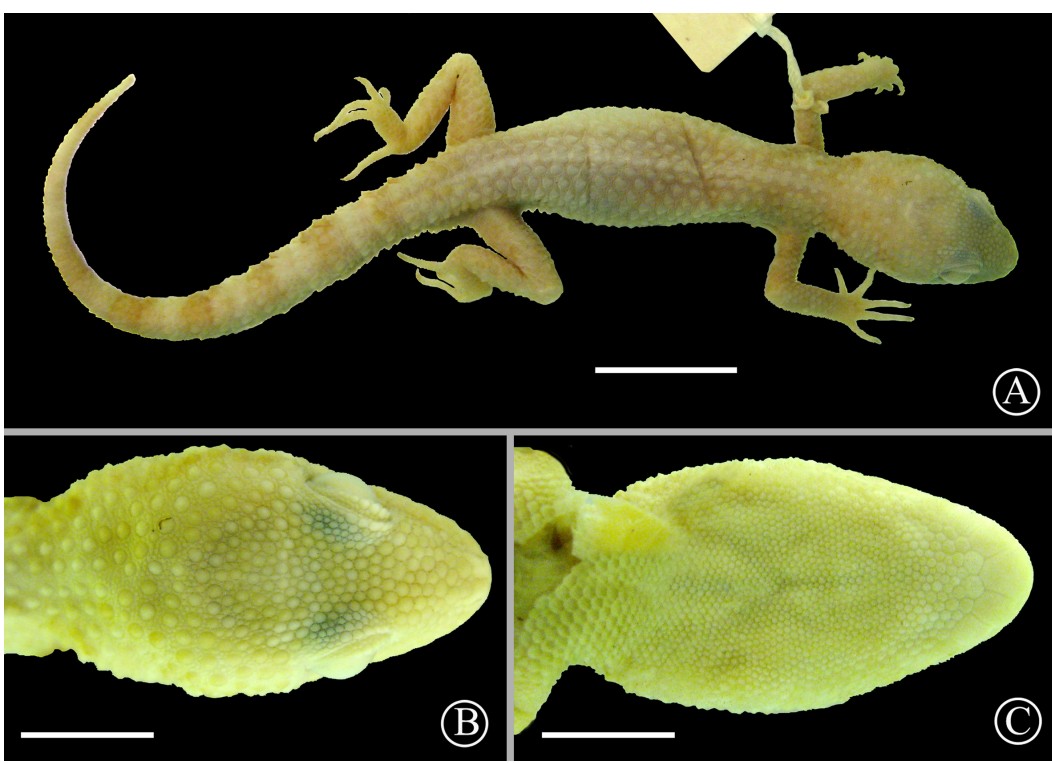

**Figure 9** **Lectotype of *Homonota horrida*.** Dorsal view (A) and details of the head in dorsal (B) and ventral (C) views of the lectotype of *Homonota horrida* (IZH-R 1). Scale bar = 10 mm (A) and 5 mm (B–C).

small granular scales; ventral scales cycloid and imbricated arranged in 18 longitudinal rows at midbody; suprascapular, axillary, and inguinal regions surrounded by small imbricated granules; sides of cloacal opening with two to three conical tubercular scales; anterior and dorsal surfaces of limbs covered by imbricated scales, slightly keeled on the dorsal surface; posterior region of limbs covered by small juxtaposed granules; ventral surface of forelimbs with juxtaposed granules, and ventral surface of hind limbs with large imbricated scales; subdigital lamellae of hands starting from pollex were recorded as follows: $8/8 - 12/12 - 14/14 - 16/16 - 8/11$; subdigital lamellae of feet starting from hallux were recorded as follow: $17/17 - 21/18 - 17/17 - 13/13 - 7/8$; large imbricated keeled scales around the tail disposed in rings, separated by two to three series of small scales.

*Coloration in preservative of the lectotype*: The specimen is at least 147 years old, and coloration is faded in most parts of the animal. The whole body is basically Cream White (52) with vestiges of blotches on the scapular region, pre and postocular lines, and rings around the tail of Salmon Color (58).

*Variation*: (Based on specimens referred in Appendix S5) SVL 42–64 mm; TrL 16–29 mm (36.9–46.0% of SVL in females, 35.7–46.8% in males); FL 7–11 mm (9.5 ± 0.30) in males, 8–12 mm (10.4 ± 0.41) in females; TL 8.3–11.4 mm (9.7 ± 0.28) in males, 8.3–12.5 mm (10.4 ± 0.35) in females; AL 11.9–14.7 mm (13.3 ± 0.38) in males, 18.8–16.8 mm (13.5 ±0.48) in females; HL 10.5–16.1 mm (12.5 ± 0.73) in males, 9.8–14.6 mm (12.7 ± 0.49) in females; HW 8.2–12.4 mm (65.2–85.5% of HL in females, 77.8–99.0% in

males); HH 4.9–7.8 mm (44.0–62.2% of HL in females, 46.2–55.2% in males); END 2.9–5.0 mm (29.6–40.0% of HL in females, 29.9–34.1% in males); ESD 3.6–6.6 mm (36.7–46.7% of HL in females, 39.0–43.9% in males); EMD 4.2–6.5 mm (35.2–47.9% of HL in females, 38.5–41.9% in males); ID 3.8–5.8 mm (29.7–54.1% of HL in females, 31.7–42.8% in males); IND 1.2–2.3 mm (11.3–23.5% of HL in females, 12.5–17.1% in males); SL 7–9; one or two elongated tubercular scales on the mouth commissure; upper region of the muzzle usually flattened, rarely slightly convex (LJAMM-CNP 6520); auditory meatus with one large scale on the upper border; IL 6–8; 13–20 longitudinal rows of ventral scales at midbody.

The coloration pattern (lost in the type series) consist of a dark and clear reticulation on the dorsal surface of the head, a dark longitudinal stripe from the tip of the snout across the temporal region extending posteriorly and upwards reaching the nuchal region. Dorsal background color usually dark with whitish transversal bands connected with a vertebral stripe of the same color. Limbs with an irregular reticulation. Ventral region of head and body always immaculate clear. Tail with dark and clear rings that can be present only on the dorsal and lateral areas of the organ, or continued to the ventral surface. Some melanic specimens (LJAM-CNP 6532, 6968) lack the vertebral stripe, and the clear transversal bands are inconspicuous.

*Distribution*: As mentioned before, this is a species complex which needs further analyses. As currently recognized, this clade is distributed from the Argentinean Province of Rio Negro in southern Argentina, to the center of Paraguayan Chaco, according to *Morando et al. (2014)*. Our analyzed samples came from Low Monte ecoregion in southern Argentina.

### *Homonota septentrionalis* n. sp.
LSID: urn:lsid:zoobank.org:act:8AE7D2A8-0D62-4AF2-8CB9-3D4346F63B52

*Holotype*: MNHNP 12238 (original field number PCS 200), adult female (Fig. 10), collected on 10 December 2014 by P. Cacciali, at Fortín Mayor Infante Rivarola (21.679°S, 62.401°W, 277 masl), Boquerón Department, Paraguay.

*Paratypes*: MNHNP 2821, 9037–8, 9131, 11406*, 11409*, 11410, 11419, 11421, 11423 (Parque Nacional Teniente Enciso, Boquerón Department, Paraguay; 21.209°S, 61.655°W, 253 masl); MNHNP 11850, 11855, 11860, 11872, 11873* (Cruce San Miguel, in front of Parque Nacional Teniente Enciso, Boquerón Department, Paraguay; 21.203°S, 61.662°W, 254 masl); SMF 101984* (topotype); SMF 29277 (Villamontes, Tarija Department, Bolivia; 21.266°S, 63.451°W, 398 masl). Holotype and specimens marked with an asterisk (*) were used for molecular analyses.

*Etymology*: The specific name *septentrionalis* is Latin, meaning "northern" and refers to the fact that this species has the northernmost distribution of all the *Homonota* species.

*Diagnosis*: This is the largest species of the genus (max. 65 mm SVL) with robust body, prominent keeled tubercles disposed in four to eight longitudinal rows, and coloration pattern of dark background with one vertebral and six or seven transversal clear bands. It can be distinguished from *H. andicola*, *H. whitii*, and *H. underwoodi* by the presence of strongly keeled dorsal scales (vs. smooth dorsal scales in *H. andicola*, *H. whitii*, and *H.*

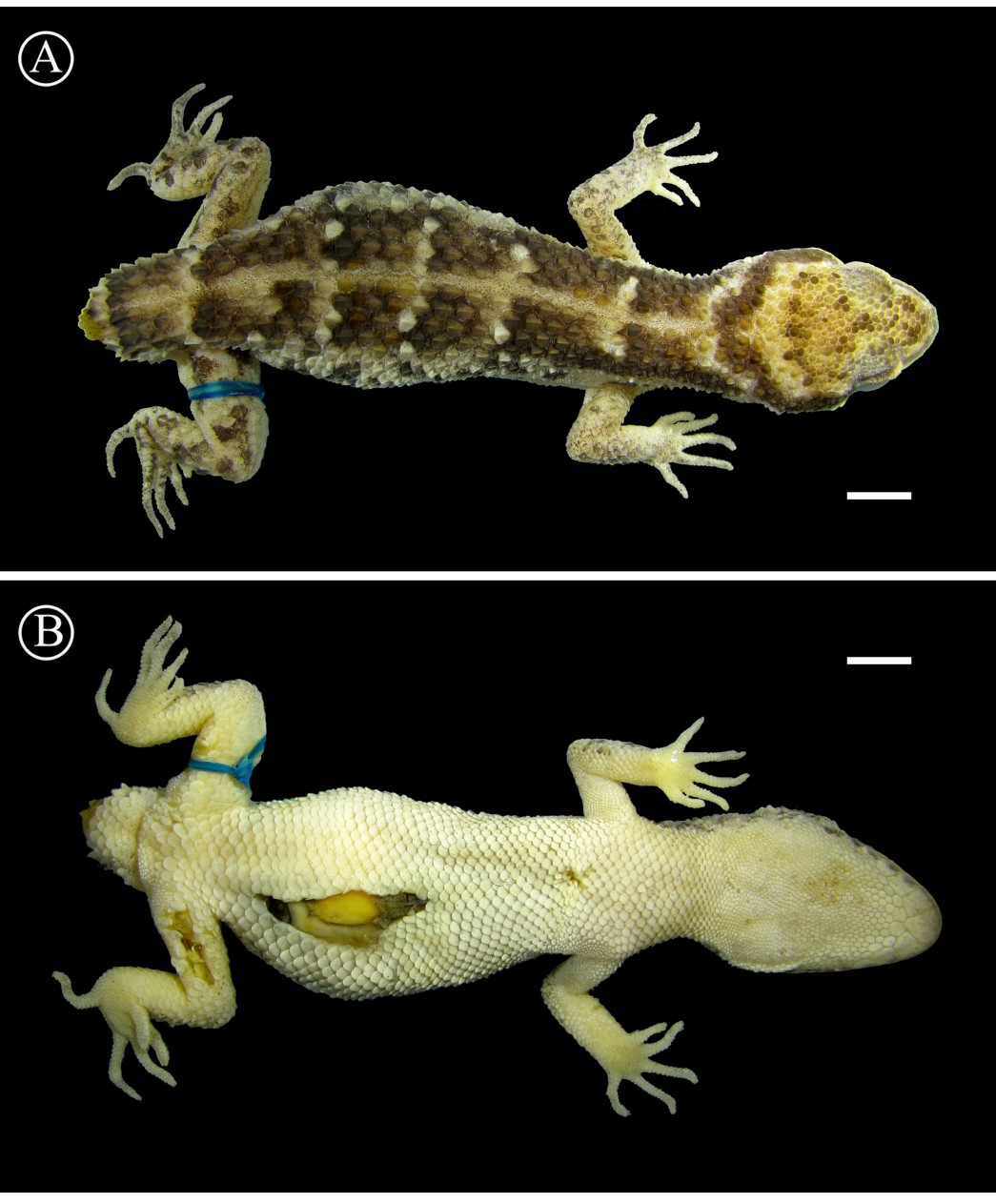

**Figure 10** **Holotype of *Homonota septentrionalis*** Dorsal (A) and ventral (B) views of the holotype of *Homonota septentrionalis* (MNHNP 12238). Scale bar = 5 mm.

*underwoodi*), transversal clear bands on a darker dorsum (vs. reticulated pattern), and from *H. underwoodi* also by a lower number of 4TL (16–20) and 3FL (11–15) (vs. 20–25 and 15–17 respectively in *H. underwoodi*). From *H. borellii* and *H. rupicola* by the oblique shape of the auditory meatus (vs. round in *H. borellii* and *H. rupicola*), transversal clear bands on a darker dorsum (vs. reticulated pattern), and also from *H. borelli* by the presence of strongly keeled dorsal scales (vs. moderately keeled), and from *H. rupicola* by a higher number of 4TL (16–20) (vs. 14–15). From *H. darwinii* by the presence of strongly keeled

dorsal scales (vs. smooth at least on the anterior part of the dorsum in *H. darwinii*), and by transversal clear bands on a darker dorsum (vs. reticulated pattern). From *H. rupicola* and *H. taragui* by the presence of enlarged keeled tubercles on the sides of the head behind the orbits (vs. homogeneous granular scales). From *H. uruguayensis* by a higher number of IL scales (6–7, vs. 4–5 in *H. uruguayensis*), by the coloration, and by the serrated edge of the auditory meatus (vs. smooth granular edge in *H. uruguayensis*). From *H. williamsii* by the presence of strongly keeled dorsal scales (vs. moderately keeled) and by transversal clear bands on a darker dorsum (vs. reticulated pattern). From *H. horrida* (the most similar species) by the high position of the auditory meatus relative to the mouth commissure (vs. lower position in *H. horrida*) (Fig. 11); less developed tubercles on the sides of the head, including a narrow area between the orbit and the auditory meatus covered with small granular scales with without or with few tubercles (vs. several big tubercles on the sides of the head even in the area between the orbit and the auditory meatus) (Fig. 11).

*Description of the holotype*: Adult female, SVL 60 mm, TrL 26 mm, tail broken near the base, FL 11.0 mm, TL 10.8 mm, AL 14.1 mm, HL 14.8 mm, HW 13.3 mm, HH 7.9 mm, END 4.6 mm, ESD 6.6 mm, EMD 5.1 mm, ID 5.5 mm, IND 2.5 mm; rostral wide with a median groove at the upper half; nares surrounded by rostral (slight contact), supranasal, two postnasals, and first SL (slight contact); SL 9/8; two elongated tubercular scales on the mouth commissure; upper region of the muzzle slightly convex covered by big homogeneous juxtaposed scales; upper surface of the head covered with big homogeneous juxtaposed scales intermixed with small granules; superciliary scales imbricated forming a serrated edge, associated to spiny-like scales on the posterior half of the orbit; lateral sides of the head heterogeneously covered with large keeled tubercles and small granular (sometimes elongated) scales; auditory meatus oblique and with serrated edge, and two big scales on the upper border; IL 6/6; mental triangular; postmentals big (less than twice the size of the following posterior scales) contacting the mental, the first IL, and a row of six posterior scales (the two centrals smaller); scales under the head reducing in size posteriorly; dorsolateral parts of the neck with granular juxtaposed scales mixed with tubercles; throat region covered by imbricated cycloid scales; dorsum covered with eight strongly keeled scales separated by one or two small granular scales, except on the vertebral area where keeled scales are separated by four granules; ventral scales cycloid and imbricated arranged in 20 longitudinal rows at midbody; suprascapular, axillary, and inguinal regions and cloacal opening surrounded by small imbricated granules; anterior and dorsal surfaces of limbs covered by large imbricated scales, keeled on the dorsal surface; posterior region of limbs covered by small juxtaposed granules; ventral surface of forelimbs with juxtaposed granules, and ventral surface of hind limbs with large imbricated scales; subdigital lamellae of hands starting from pollex were recorded as follows: $7/8 - 12/10 - 13/14 - 13/13 - 12/10$; subdigital lamellae of feet starting from hallux were recorded as follow: $13/13 - 18/18 - 15/14 - 12/12 - 10/10$; large imbricated scales around the tail (stump) with the eight uppermost strongly keeled.

*Coloration in life*: Dorsal surface of head Grayish Horn Color (268) with groups of Dusky Brown (285) scales, irregularly mixed with Hair Brown (277) scales; posterior surface of the head with a curved Hair Brown (277) line interrupted by five groups of

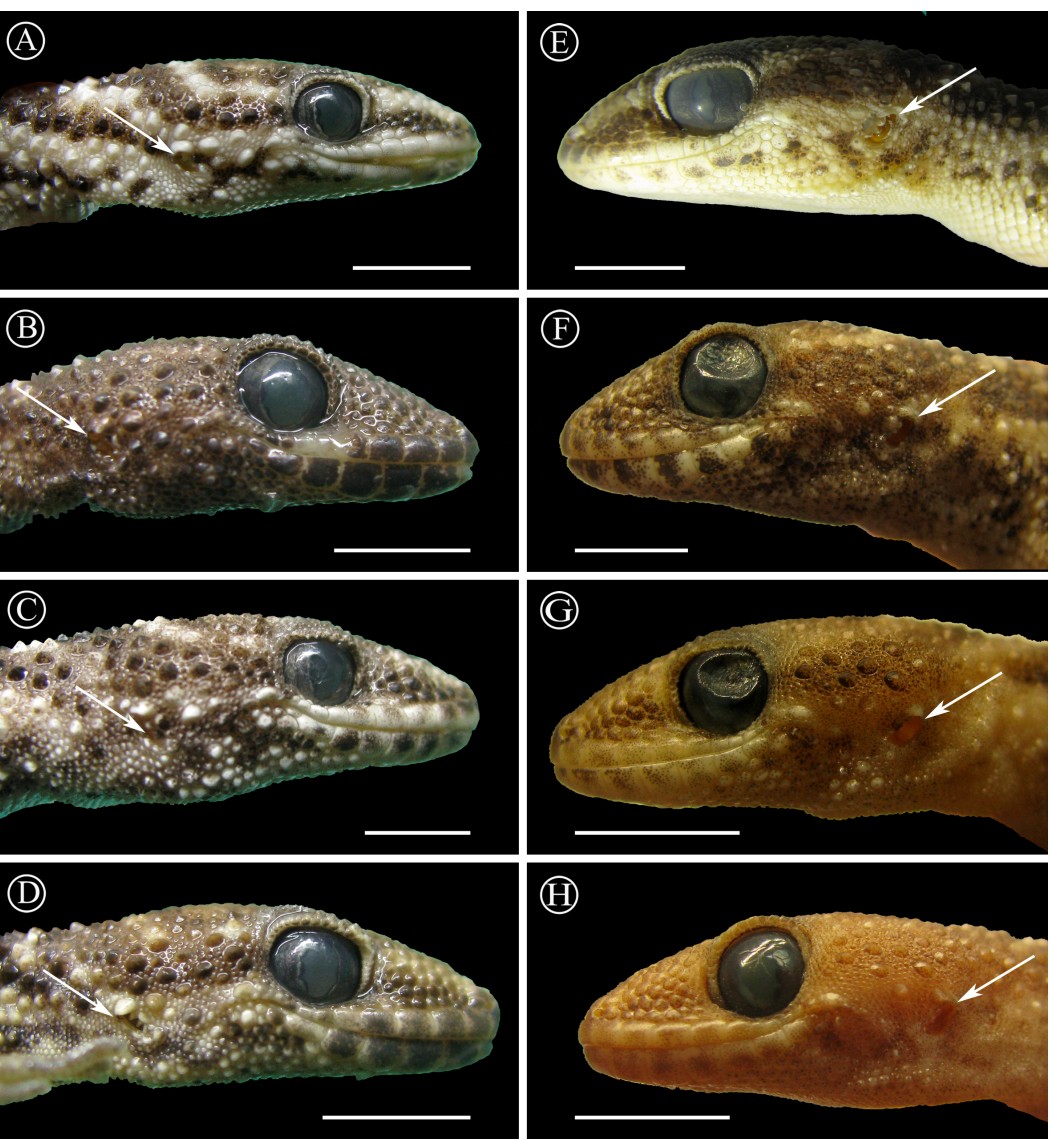

**Figure 11** **Position of ear opening.** Lateral sides of the head of *Homonota horrida* (A–D) compared with *H. septentrionalis* (E–H) showing differences in the disposition of ear opening (EO), indicated with white arrows, and the tubercles between the EO and the commissure of the mouth. Vouchers: LJAMM-CNP 6520, 6532, 6533, 7670 from A to D respectively, and MNHNP 12238, 11855, 11406, 9131 from E to H respectively. Scale bars = 5 mm.

Dusky Brown (285) scales; upper lateral view of the head Grayish Horn Color (268), edged below by a thick Dusky Brown (285) stripe from the muzzle (interrupted by the orbit) to the temporal region; supralabial and infralabial regions Smoky White (261) with irregular Raw Umber (280) suffusions on the 1st and 2nd SL and 1st to 5th IL; region between mouth commissure and shoulder Smoky White (261) with irregular Dusky Brown (285) speckles, edged above (bordering the upper edge of the ear opening) by an irregular Cream Yellow (82) stripe; ventral surface of the head Smoky White (261); dorsal ground color Dusky Brown (285), with a Light Straw Yellow (95) vertebral stripe, and five transversal

Light Sulphur Yellow (93) lines; lateral parts of the body Cream Yellow (82) with irregular Dusky Brown (285) speckles; venter Smoky White (261); dorsal surface of limbs Cream Color (12) with irregular Dusky Brown (285) speckles on the forelimbs, and groups of Dusky Brown (285) scales (eventually forming short stripes) on the hind limbs; ventral surface of limbs Smoky White (261).

*Coloration in preservative*: Dorsal surface of head Drab (19) with groups of Vandyke Brown (282) scales; posterior surface of the head with a curved Vandyke Brown (282) line; upper lateral view of the head Smoke Gray (266), edged below by a thick Raw Umber (260) stripe from the muzzle (interrupted by the orbit) to the temporal region; supralabial and infralabial regions Cream White (52) with irregular Raw Umber (260) suffusions on the 1st and 2nd SL and 1st to 5th IL; region between mouth commissure and shoulder Cream White (52) with irregular Raw Umber (260) speckles; ventral surface of the head Cream White (52); dorsal ground color Raw Umber (260), with a Beige (254) vertebral stripe, and five transversal Cream White (52) lines; lateral parts of the body Cream White (52) with irregular Raw Umber (260) speckles; venter Cream White (52); dorsal surface of limbs Beige (254) with irregular Sepia (279) speckles on the forelimbs, and groups of Sepia (279) scales (eventually forming short stripes) on the hind limbs; ventral surface of limbs Cream White (52).

*Variation*: SVL 37–65 mm; TrL 15–28 mm (43.3–48.2% of SVL in females, 38.3–48.8% in males); Tail length 47–63 mm (ratio SVL:Tail - 1:1 in one female, 1:1.18–1:1.22 in two males, and 1:1.17 in a juvenile of unknown sex); FL 8–9 mm (8.8 ± 0.37) in males, 10–12 mm (11.2 ± 0.83) in females; TL 7.2–9.8 mm (8.7 ± 0.36) in males, 9.4–11.3 mm (10.5 ± 0.81) in females; AL 10.2–13.1 mm (11.7 ± 0.91) in males, 13.1–15.0 mm (14.1 ± 0.76) in females; HL 10.7–13.3 mm (11.8 ± 0.38) in males, 12.9–17.3 mm (14.6 ± 1.66) in females; HW 8.1–13.3 mm (71.6–89.8% of HL in females, 75.7–84.4% in males); HH 5.8–8.6 mm (49.7–61.3% of HL in females, 54.1–61.4% in males); END 3.7–5.8 mm (31.9–37.9% of HL in females, 29.3–39.1% in males); ESD 3.6–6.8 mm (39.3–46.7% of HL in females, 31.6–45.9% in males); EMD 3.6–5.6 mm (34.4–40.8% of HL in females, 33.0–38.6% in males); ID 3.7–5.5 mm (30.1–38.7% of HL in females, 33.0–38.3% in males); IND 1.4–2.5 mm (14.4–16.9% of HL in females, 12.3–18.8% in males); SL 6–9; one or two elongated tubercular scales on the mouth commissure; upper region of the muzzle slightly convex or flattened; auditory meatus with one or two big scales on the upper border; IL 6–7; 12–20 longitudinal rows of ventral scales at midbody.

The coloration variation follows the same pattern observed for the holotype. Smaller animals (MNHNP 11419, 11423) are clearer and the clear transversal bands are reduced to the paravertebral area; vertebral stripe reduced in MNHNP 11855; three paratypes (MNHNP 2821, 9037, 9131) have a darker pattern being reddish dorsal background color, and in two of them (MNHNP 2821, 9131) the transversal bands are almost faded; the original tail (MNHNP 9131, 11419, 11421, 11850, 11860, 11872, SMF 29277) has transversal dark and clear bands dorsally, and clear or reddish hue ventrally.

*Distribution*: *Homonota septentrionalis* is distributed in the northernmost range of the genus. The examined specimens come from the Dry Chaco, at the westernmost part of the Paraguayan Chaco and southeast of Bolivia (Fig. 12).

*Habitat*: The environment inhabited by *H. septentrionalis* is a xerophytic (precipitation varies between 300 and 400 mm per year) and thorny dry forest, with null or scarce herbaceous stratum (Fig. 13). This species is a nocturnal ground dweller, being abundant in natural areas, and also present in anthropogenically modified areas.

## DISCUSSION

The analysis of genetic barcodes of the mtDNA gene 16S provided the first evidence for the existence of an undescribed species of *Homonota* in Paraguay, which was posteriorly tested with additional data. The uncorrected genetic distance of the 16S fragment between *H. horrida* and *H. septentrionalis* is rather low (1.8–2.5%) compared to distances between species of other genera of geckos such as *Diplodactylus* (4–12%; *Pepper, Doughty & Keogh, 2006*), *Phyllopezus* (6–15%; *Gamble et al., 2012*), and *Lepidoblepharis* (12–23%; *Batista et al., 2015*). Using the species delimitation program ABGD, we estimated the intraspecific variation since this program explores the pairwise differences in barcode datasets, providing limits for intraspecific divergence (*Puillandre et al., 2012*). The expected intraspecific variation for *Homonota* Species A and Species B, matches with the variation in uncorrected pairwise distance (Table 1), with a clear difference between the two taxa. The tree-based PTP analysis provides speciation models based on number of substitution in a phylogenetic hypothesis, for which the branch length of a tree represents the number of substitutions (*Zhang et al., 2013*). This algorithm also suggested two putative species, one from Argentina (Species A) and the other from Paraguay (Species B).

The topology of the species tree (Fig. 2) shows *Phyllodactylus* as the sister genus of *Homonota*, congruent with *Gamble et al. (2008b)*; *Gamble et al. (2011)* and *Morando et al. (2014)*. The arrangement among groups of *Homonota* inferred the *fasciata* group as the most basal clade, a hypothesis contrary to that proposed by *Morando et al. (2014)* where the *whitii* group was the most basal clade within *Homonota*. The majority of the topological arrangements among the concatenated trees are identical, with the exception of the position of *H. taragui* which was closely related to *H. rupicola* using mitochondrial genes, and related to *H. borellii* using nuclear genes (Appendix S9); a conflict that was already reported by *Morando et al. (2014)*. In our phylogeny *H. horrida* and *H. septentrionalis* were inferred to as sister taxa with high statistical support ($PP = 1$, Fig. 2). Given the taxonomic modifications proposed here, we suggest referring to the group that contains *H. underwoodi*, *H. horrida*, and *H. septentrionalis* as the *H. horrida* species group.

The holotype of *Homonota fasciata* was sent to Paris by Auguste Plée who was a botanist who collected several samples of plants and animals in the Antilles, and some of his collections are valid records for Martinique (i.e., type locality of *H. fasciata*) such as *Monstera adansonii* (Alismatales: Araceae), *Auxis thazard* (Actinopterygii: Scombridae), *Eleutherodactylus martinicensis* (Amphibia: Eleutherodactylidae), *Mabuya mabouya* (Reptilia: Scincidae), *Megalomys desmarestii* (Mammalia: Cricetidae), whereas some others were recorded but currently extinct as *Leptodactylus fallax* (Amphibia: Leptodactylidae) and *Leiocephalus herminieri* (Reptilia: Leiocephalidae) (*Madison, 1977*; *Collette & Aadland, 1996*; *Borroto-Páez & García, 2012*; *Hedges & Conn, 2012*; *Breuil, 2015*). Thus, although

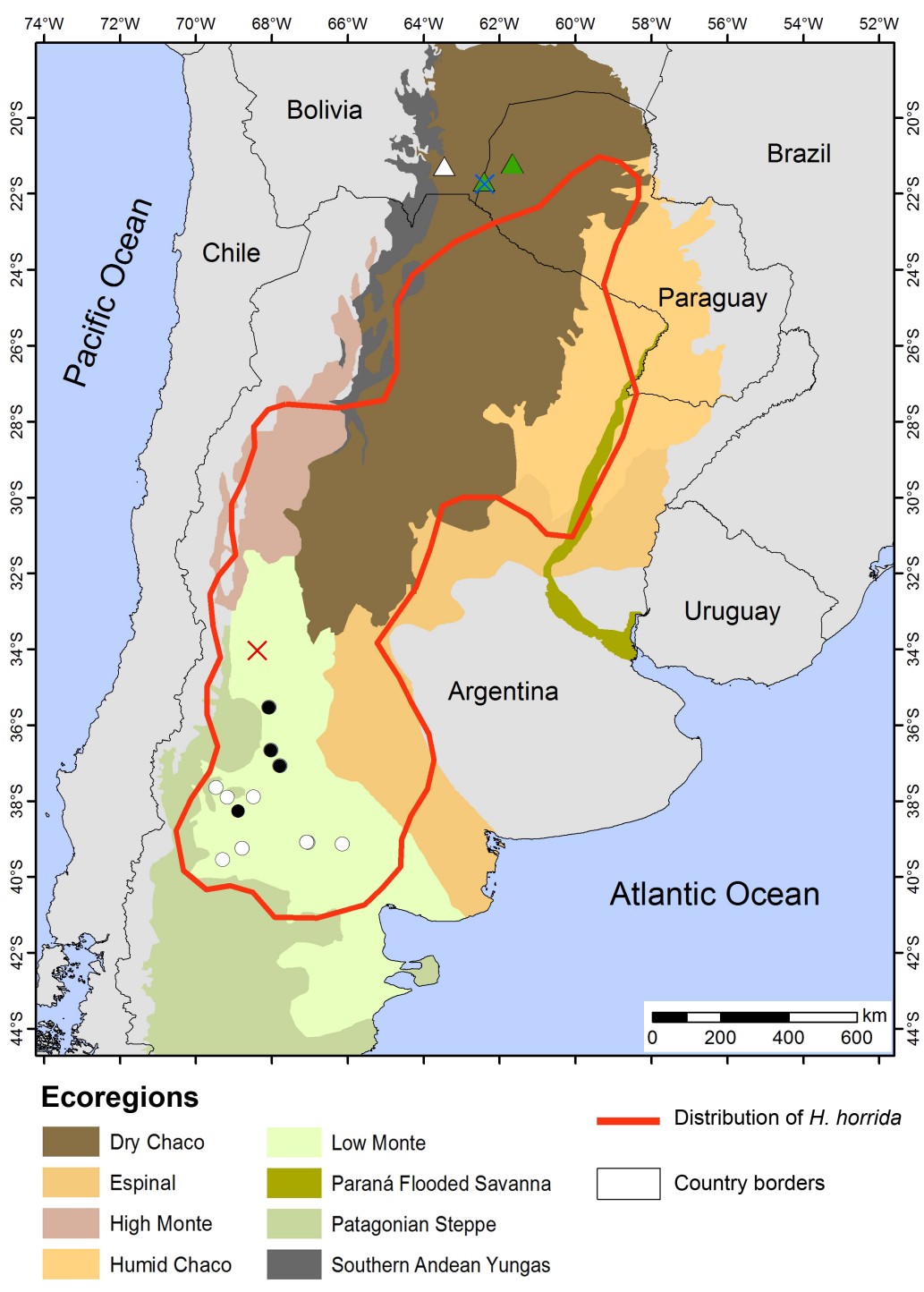

**Figure 12  Distribution Map.** Locality records of *Homonota septentrionalis* (triangles) highlighting localities of specimens used for genetic analyses (green triangles), and the distribution of *Homonota horrida* (red line) according to *Morando et al. (2014)* with localities of specimens used for morphological analyses (white circles) and genetic analyses (black circles). Crosses represent type localities: blue for *H. septentrionalis*, and red for *H. horrida*.

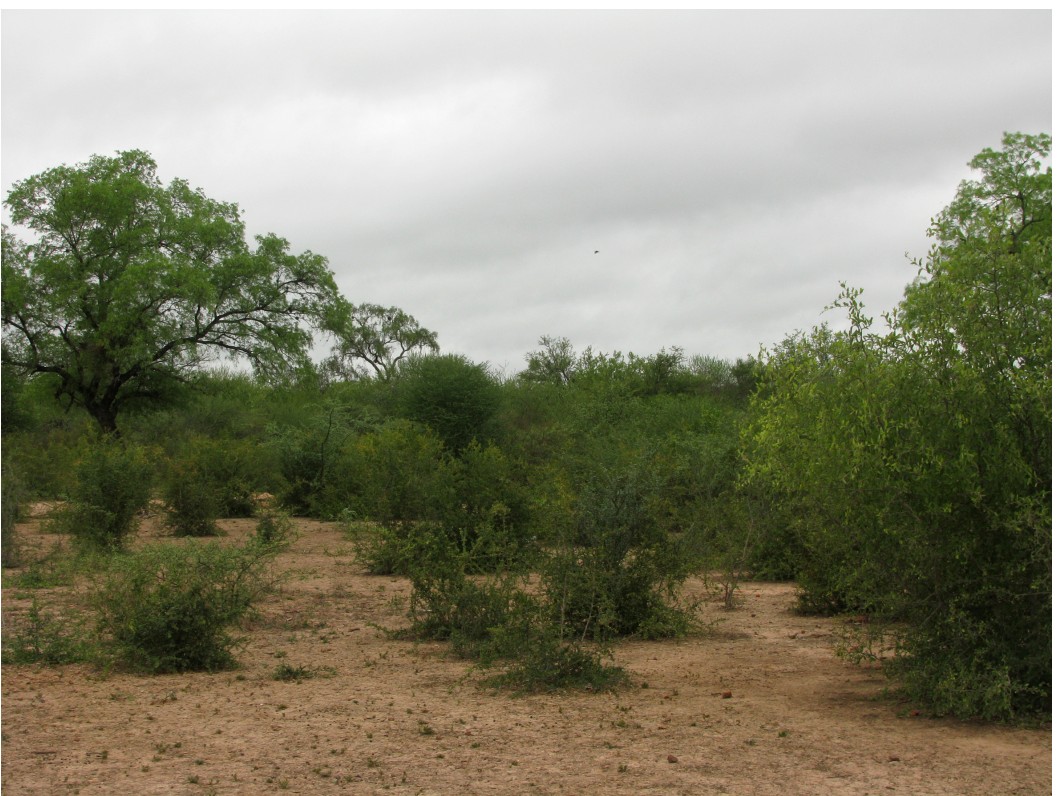

**Figure 13 Habitat of *Homonota septentrionalis*.** Environmental characteristics of the type locality of *H. septentrionalis*.

some locality records provided by Plée are trustable, the name *H. fasciata* based on specimen MNHN 6756, remains has to be considered as a *species inquirenda*. More historical analyses could shine some light on the real origin of this specimen.

*Abdala & Lavilla (1993)* stated that differences between *Homonota horrida* and the type of *H. fasciata* were due to variation, which is true for some meristic characters. Nevertheless, the small size of postmental scales and serrated edge of auditory meatus are common morphological traits of *H. horrida*. These authors suggested that some specimens of *H. horrida* could have big postmentals and smooth auditory meatus (referring to specimens FML 35 and FML 114) which is rare for the species. Another common trait for *H. horrida* is the presence of a tubercular scale on the upper edge of the auditory meatus, which is absent in the type of *H. fasciata*. Further genetic and morphological analyses of Argentinean populations of *H. horrida* are required for a better understanding of variation within the species.

*Homonota septentrionalis* is a large species of *Homonota*, with a marked sexual dimorphism in measurable characters according to the DA analysis (Fig. 3), where SVL and TrL are the variables that contribute more to the differentiation (Appendix S10). This differs from what is known for *Homonota darwinii* where *Ibargüengoytía & Casalinas (2007)* found no sexual dimorphism, although *Fitch (1981)* reported differences in SVL between males and females in Gekkonidae with females usually larger than males. More analyses

are needed in order to explore the extent of this pattern in other species of the genus.

Genetic analyses were key for the recognition of the new species, since the morphological differences between *H. septentrionalis* and *H. horrida* are subtle and they could be considered cryptic species. High degree of genetic differentiation and low degree of morphological distinction is a common phenomenon for lizards, leading to situations in which authors designate candidate species without formal descriptions (*Gamble et al., 2012*; *Werneck et al., 2012*), or cases in which authors base the entire diagnosis upon genetic clustering (*Leaché & Fujita, 2010*).

Currently, *Homonota septentrionalis* is known from the type locality (Fig. 11), in plain areas and xerophytic environments. Given the similarity in external morphology between *H. septentrionalis* and *H. horrida* it is difficult to elaborate a cresonymy list of the previous records for these species. Records published by *Mendoza, Rivas & Muñoz (2015)* as *H. fasciata* from Bolivia, probably are *H. septentrionalis*, but further morphological and genetic analyses are required for a better understanding of the distribution pattern of *H. septentrionalis*.

Based on these results, the actual diversity of the genus *Homonota* is as follows: *borellii* group: *H. borellii*, *H. uruguayensis*, *H. rupicola*, and *H. taragui*; *horrida* group: *H. horrida*, *H. underwoodi*, and *H. septentrionalis* sp. nov; *whitii* group: *H. whitii*, *H. darwinii*, *H. andicola*, and *H. williamsii*; *Incertae sedis*: *H. fasciata*.

Currently, the conservation status of *Homonota septentrionalis* is totally unknown. *Homonota fasciata* was categorized as Least Concern (LC) by *Motte et al. (2009)* given its big range, but since we actually do not know the range of *H. septentrionalis*, the conservation status might be different. This species is related to the Dry Chaco, which for a long time was a sanctuary for wildlife because of the lack of anthropogenic impacts; but unfortunately in the last decade the deforestation is severely threatening many areas of the Dry Chaco (*Eva et al., 2004*; *Caballero et al., 2014*). An assessment of the status of this new taxon is required.

## ACKNOWLEDGEMENTS

We thank Emilio Buongermini, Ignacio Ávila, Norman Scott, and Aníbal Bogado for help during fieldwork. Also to Cristian Pérez (LJAMM-CNP), Nicolás Martínez (MNHNP), and Linda Mogk (SMF) for assistance during revision of specimens in scientific collections. We are grateful to the staff (especially Heike Kappes) of the Grunelius-Möllgaard Laboratory (SMF), and Joao Diogo (Lab Center, BiKF, SMF) for lab support, and to Diego Barrasso (Instituto de Diversidad y Evolución Austral, Chubut, Argentina) for comments on software implementation. We also acknowledge Nicolás Vidal and staff of the MNHN for lending material under their care, and Karla Schneider (IZH) who let us analyze specimens under her care.

## Funding

PC and MMotte received financial support for fieldwork activities from Consejo Nacional de Ciencia y Tecnología (Paraguay), through PRONII (Programa Nacional de Incentivo a los Investigadores), and field equipment was supplied by Idea Wild. PC also received financial support for revision of scientific collections from Amerisur Paraguay. This work is part of an ongoing project of Barcoding of the Paraguayan Herpetofauna, as part of the PhD work of PC, funded by the Deutscher Akademischer Austauschdienst (DAAD, Germany). The publication of this article was funded by the Open Access Fund of the Leibniz Association. The funders had no role in study design, data collection and analysis, decision to publish, or preparation of the manuscript.

## Grant Disclosures

The following grant information was disclosed by the authors:
PRONII (Programa Nacional de Incentivo a los Investigadores).
Amerisur Paraguay.
Deutscher Akademischer Austauschdienst (DAAD, Germany).
Open Access Fund of the Leibniz Association.

## Competing Interests

The authors declare there are no competing interests.

## Author Contributions

- Pier Cacciali conceived and designed the experiments, performed the experiments, analyzed the data, wrote the paper, prepared figures and/or tables.
- Mariana Morando conceived and designed the experiments, performed the experiments, analyzed the data, reviewed drafts of the paper.
- Cintia D. Medina performed the experiments, analyzed the data, prepared figures and/or tables, reviewed drafts of the paper.
- Gunther Köhler conceived and designed the experiments, contributed reagents/materials/analysis tools, reviewed drafts of the paper.
- Martha Motte performed the experiments, reviewed drafts of the paper.
- Luciano J. Avila conceived and designed the experiments, performed the experiments, contributed reagents/materials/analysis tools, reviewed drafts of the paper.

## Animal Ethics

The following information was supplied relating to ethical approvals (i.e., approving body and any reference numbers):

Collecting permits SEAM No 04/11 and SEAM No 133/2015 were issued by the Secretaría del Ambiente in Paraguay.

## DNA Deposition

The following information was supplied regarding the deposition of DNA sequences:
MF278828–MF278854.
## Data Availability

Raw data are available as Appendices in Supplemental Information 1. Raw data for MF535517 are available as Supplemental Information 2.

## New Species Registration

The following information was supplied regarding the registration of a newly described species:

Genus name: urn:lsid:zoobank.org:act:22AF067B-1B91-4736-AE2E-779B97BF1F23
Publication: urn:lsid:zoobank.org:pub:04A88748-40CA-4243-BF28-96B67A646E35
Publication: urn:lsid:zoobank.org:pub:7233E738-D8B3-424D-B1FC-7CA903BED5A0
Species name: urn:lsid:zoobank.org:act:8AE7D2A8-0D62-4AF2-8CB9-3D4346F63B52.

## Supplemental Information

Supplemental information for this article can be found online at http://dx.doi.org/10.7717/peerj.3523#supplemental-information.

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
