# Peer review of "Taxonomic analysis of Paraguayan samples of Homonota fasciata Duméril & Bibron (1836) with the revalidation of Homonota horrida Burmeister (1861) (Reptilia: Squamata: Phyllodactylidae) and the description of a new species"

_PeerJ, doi:10.7717/peerj.3523_

## Round 0.1 · original submission · Major Revisions

Both reviewers found value in your manuscript and in the problem that you are addressing. However, there are some deficiencies that will have to be addressed before the manuscript can be accepted for publication.

I have read and carefully considered the reviewers' suggestions and I find them all to be reasonable and pertinent. In particular, please pay special attention to the following points raised by Reviewer 1 in revising the manuscript:

1. The map is missing crucial information relevant to your hypothesis.

2. Inadequate documentation of the specimens used in the phylogenetic analyses, and their localities.

3. Consider revisiting the phylogenetics to perform a combined analysis. Also, since a test of species delimitation exists, it would be wise to use it, or present a compelling reason not to.

4. More clarity is needed in the text regarding names - it is often difficult or impossible to determine which taxon is being discussed.

This list is hardly exhaustive - I expect you to address all of the reviewers' concerns.

Please take these comments in the constructive spirit in which they are intended. I look forward to seeing an improved version of this work in the near future. Best of luck with the revisions.

Reviewer 1 ·

Basic reporting

1. Basic Reporting
-The article must be written in English using clear and unambiguous text and must conform to professional standards of courtesy and expression.
The article is well written, using clear text in most of it, however there is ambiguity in the subtitles of figures 6, 7 and 8, when the authors refer to a "banded, large-scale Homonota" without indicate the species and voucher number of specimen in the photograph.


-The article should include sufficient introduction and background to demonstrate how the work fits into the broader field of knowledge. Relevant prior literature should be appropriately referenced.
The authors have a significant and interesting taxonomic problem in hand, specially regarding Homonota fasciata which certainly deserves the initiative, justifying the study. The introduction is well written, providing an adequate background on taxonomic history and presenting the questions with will be dealing in the ms.


- The structure of the submitted article should conform to an acceptable format of ‘standard sections’ (see our Instructions for Authors for our suggested format). Significant departures in structure should be made only if they significantly improve clarity or conform to a discipline-specific custom.
The ms follow PeerJ guidelines, however lacks the section "Conclusions".


-Figures should be relevant to the content of the article, of sufficient resolution, and appropriately described and labeled.
In general, the figures are relevant for the questions tackled, specifically the removal of H. horrida from sinonimy and the description of H. septentrionalis. However, The distribution map lacks essential information to defend the hypothesis that H. septentrionalis represents a new species. The distribution of H. horrida, as proposed by the authors, is not showed, there is only 3 points representing the distribution of H. septentrionalis. This is an issue, since one of the arguments used by the authors to propose the new species is the "allopatry and different biogeographic regions" (line 193). Another information that are not represented in the map is the localities where the authors have only molecular information, only morphological info, both, or records from literature. The authors should also identify the localities represented in the map on the subtitles, using numbers or letters. The absence of these info strongly hamper the interpretation of results. Additionally, as noted above, there are problems with subtitles of figures and tables. See the annotations on pdf file for more details.


-The submission should be ‘self-contained,’ should represent an appropriate ‘unit of publication’, and should include all results relevant to the hypothesis. Coherent bodies of work should not be inappropriately subdivided merely to increase publication count.
No problems here.

Experimental design

-The submission must describe original primary research within the Aims & Scope of the Journal.
The ms deals with an interesting and original research, however, there are problems with the soundness of data (see below).

-The submission should clearly define the research question, which must be relevant and meaningful. The knowledge gap being investigated should be identified, and statements should be made as to how the study contributes to filling that gap.
The questions dealt in the manuscript are well defined, the taxonomic status of Paraguayan populations of Homonota.

-The investigation must have been conducted rigorously and to a high technical standard.
No problems here.

-Methods should be described with sufficient information to be reproducible by another investigator.
There are serious problems in this issue, mostly regarding specimen and samples info. First, specimen info used in both phylogenetic analysis (16s and 11 genes) is lacking on the tables, as locality info and coordinates. For example, the coordinates of H. horrida specimens analysed are missing in appendix S2, and locality and coordinates from samples sequenced for 11 genes are also missing in appendix S1. There is no table showing the samples´info used in 16s analysis; some of this info could be recovered from the species description (a hard task: why not make reader´s work easier?), but not all the info (I was unable to find locality info for MNHNP 11409 and 11873, specimens used in the phylogenetic analysis to advocate the species status of H. septentrionalis, but not mentioned on the species description). All these problems hamper the repeatability of your work, e.g. if one download your sequences from GenBank for future analysis. I suggest a major revision on info about specimens used in molecular and morphological analysis, and also in the distribution map, as noted above. Another problem is the manual editions on alignments performed, which should be avoided, even if it represents a common practice in the literature (see comments on pdf file for more details and suggestions to deal with it).
Another problem is the myriad of names used in the ms to refer to species H. horrida, H. fasciata and specially for H. septentrionalis. I made a list of names used. Sometimes is impossible to have sure which taxon the authors are referring to. Here is the list, in order of appearance in the text:
-"banded, large-scaled Homonota"
-"paraguayan pops. of H. fasciata"
-"Homonota sp. "Paraguay""
-In 11 genes analysis: "Homonota fasciata" in the trees refers to H. horrida?? This is not a conclusion of the paper? Why leave the name fasciata? This also apply for when referring to H. fasciata from Mendoza (e.g. fig 1)
-"H. fasciata common usage"
-"H. horrida"
-"H. septentrionalis"
-"Homonota sp." (fig. 2, 3 and 4)
-"H_fas_ss" and "H_aff_fas" (figs 3 and 4)
I suggest rewrite the ms in order to make clear of which taxon the authors are refering to, and get rid of several of these names. This will facilitate readers´job to understand this already confusing taxonomic situation.

- The research must have been conducted in conformity with the prevailing ethical standards in the field.
The authors provided the numbers of collecting permits issued by Paraguayan government. However it is a good practice to detail how the vouchers were euthanized.

Validity of the findings

-The data should be robust, statistically sound, and controlled.
In both morphological and molecular analyses, the sample siza is small, which hampers the conclusions of the ms (more details below).

-The data on which the conclusions are based must be provided or made available in an acceptable discipline-specific repository.
No problem here.

-The conclusions should be appropriately stated, should be connected to the original question investigated, and should be limited to those supported by the results.
The ms have three main conclusions: remove H. horrida from sinonimy with H. fasciata, designate H. fasciata as species inquirenda and apply a specific status for Homonota from Paraguay, H. septentrionalis. In my opinion, the authors made the case for the former 2 conclusions, but the evidence for give an specific status to Paraguayan pops. are weak. First, the morphological difference between H. horrida and H. septentrionalis is very subtle, as noted by the authors (line 523). Regarding DA analysis (continuous and discrete variables), the 95% CI plot only separate the females of H. septentrionalis in continuous variables, but even so with a considerable overlap (inset of figure 3). This is manly due to small sample sizes, as also noted by the authors (line 220). Here a parenthesis: the authors did not clearly stated which variables are responsible for the differentiation between two taxa. I supposed that SVL is responsible for the great percentage of variation explained by first axis in both analysis. This is important, as could be a diagnosis character between taxa. Also, differences in temporal tubercles and disposition of ear opening are also subtle (MNHNP 12238 and 11855 do present tubercles between the comissure of mouth and ear opening (fig. 11); the diagnosis "less developed tubercles on the sides of the head, including a narrow area between the orbit and the auditory meatus covered with small granular scales with without or with few tubercles (vs. several big tubercles on the sides of the head even in the area between the orbit and the auditory meatus)" (lines 381-384) are very abstract.
This problems could have less importance if the molecular evidence was robust, which is not the case. The genetic distance on 16s between H. horrida and H. septentrionalis is low (~2%), as noted by the authors (line 476). Soon after, the authors stated that genetic distance between two taxa in cyt-b is high (14%), however, the authors did not explain why such a difference between two measures in the same genome. But the main problem, as I see it, is that the dataset used in 16s analysis is distinct from the 11 genes. If I get it right (due to absence of specimen info), from the 3 samples used in 16s analysis, one is from Bolivia, other is from type locality (Fortín Mayor) and the last one I don´t know where it from. The 2 samples used in the more comprehensive phylogenetic analysis (11 genes), one is from Parque Nacional Teniente Enciso, and the other I also don´t know the locality. My point is: the datasets are not equal, nor from the same localities. I am afraid that could be more than one entity in the type series of H. septentrionalis, given this context. Also, molecular conclusions are based in small sample sizes (5 specimens in total). Another problem with phylogenetic analysis is that is hard to define the locality of samples from "H. fasciata" used in 11 genes analysis, which is also different from the samples used in 16s analysis (although LJAM-CNP 5047 is common to both, but only this sample, and I was not able to find the locality of this specimen).
Finally, is not clear to me why perform 2 analyses, rather than join the data in one, more robust analysis. Specially if the analyses differ in conclusions, which seems to be the case here, given differences in genetic distances. I would suggest perform a combined analysis (which certainly will involve more sequencing). Also, There is analysis of species delimitation which specifically test the hypothesis that 2 lineages are distinct species (e.g. BPP (Yang 2015), and others).

Additional comments

The research question dealt in the ms is important and relevant, and is clear that the authors made a true effort in achieve the conclusions. However, there are problems in the organization of data, especially in specimen and samples information, and in figures and tables. Also, the two molecular datasets used show somewhat conflicting results, which were not handled accordingly by the authors. Finally, I believe that the conclusions were not supported by the analyses, more specifically the description of H. septentrionalis, which should be tested by analyses specifically designed to test species delimitation. Therefore, I ask for major revisions, and I make myself available to review future versions of ms.

Annotated reviews are not available for download in order to protect the identity of reviewers who chose to remain anonymous.

·

Basic reporting

The study is a clear, concise, and methodical examination of relevant specimens of the Homonota genus that supports the recognition of a new species, re-establishment of a species name previously relegated to synonymy, and the existence of a relatively rare example of sexual dimorphism in a gecko.

The manuscript PASSES basic reporting.

Minor revisions necessary are noted on the marked-up pdf attached to the review and most important summarized here:
-the interpretation of Fitch (1981) is incorrect (see note on pdf)
-Figures and figure legends are inconsistent in formatting
-Literature cited section lacks citations of taxonomic authorities given for taxonomy. This is necessary to confirm that Article 29 of the ICZN is satisfied.
-Description of voucher for H. septentrionalis should give age class and gender of voucher.

Experimental design

PASSES. Experimental design is exhaustive in planning and execution. The molecular-based phylogenetic analyses are well described and appropriate. The morphological measures are complete and exceed most published accounts.

Points of improvement (see also marked pdf):
-justify use of outgroup in molecular analyses

Validity of the findings

PASSES.

The arguments and data clearly support the conclusions of the authors.

Additional comments

Very well done and requires only minor revisions prior to publication. Please carefully review the comments on the PDF for specific suggestions for revision.

---

## Round 0.2 · Minor Revisions

Thank you for your attention to detail in revising the manuscript. There are only a handful of improvements required before the manuscript will be acceptable for publication. In particular, please pay attention to the review comments on the figures and tables. I agree with the reviewer that adding scale bars to Figure 11 would be a straightforward and helpful upgrade.

·

Basic reporting

Seems to be a much improved submission. Minor corrections needed, but only at the editorial and not the scientific level. Figures are good, but some concerns are noted on the attached pdf. Overall I think the paper is nearly ready for publication.

Experimental design

Very good. Process results in convincing data.

Validity of the findings

Without getting into the trap of species concept debates, I find the results and interpretation to be valid and supported by good empirical evidence.

Additional comments

Very nicely done. The introduction section is exemplary and should be a model for how to present relevant background information.

Please carefully review the attached PDF for minor comments:
-Line 70: in the parenthetical statement do you mean subspecies of H. fasciata or species of the H. fasciata group?
-Line 329: change to "Diagnosis: A large species of..."
-Line 350: do you mean "axillary"?
-Line 360: no changes required. I just wonder if examination of the preserved holotype under UV light (or another ALS) would capture some pattern?
-Discussion, while possibly beyond the scope of the study it seems that the high number of species of Homonata in this region of South America could be discussed from an historical perspective. Is is explosive radiation? Are any fossils known? Not necessary from the point of view of modern diversity, but I like to see biotic patterns put into historical context.
-See pdf for comments on figures and tables and the associated captions.

---

## Round 0.3 · accepted · Accept

Thank you for your attention to detail in addressing the reviewers' concerns. I am satisfied that the revised manuscript meets the PeerJ's criteria for publication, and I am happy to accept it.

The decision of whether or not to publish the peer reviews alongside the paper is entirely yours, and will not affect how your paper is handled going forward. However, I encourage you to do so. Making the reviews public allows the reviewers to receive more credit for their efforts, and also contributes to the emerging culture of fairness and transparency in editing and peer review.